# *NeedleBench*: Evaluating LLM Retrieval and Reasoning Across Varying Information Densities

**Mo Li**                                                       *limo.research@gmail.com*
*Tsinghua University*
*Shanghai AI Laboratory*

**Songyang Zhang**[*]                                      *zhangsongyang@pjlab.org.cn*
*Shanghai AI Laboratory*

**Taolin Zhang**                                              *zhangtlin3@gmail.com*
*Tsinghua University*
*Shanghai AI Laboratory*

**Haodong Duan**                                           *duanhaodong@pjlab.org.cn*
*Shanghai AI Laboratory*

**Yunxin Liu**                                            *liuyunxin@air.tsinghua.edu.cn*
*Tsinghua University*

**Kai Chen**[*]                                                  *chenkai@pjlab.org.cn*
*Shanghai AI Laboratory*

**Reviewed on OpenReview:** *https://openreview.net/forum?id=cEvmIKsRw0*

## Abstract

The capability of large language models to handle long-context information plays a crucial role across various real-world applications. Existing methods for evaluating long-context abilities often rely either on real-world long texts, making it difficult to exclude the influence of models' inherent knowledge, or introduce large amounts of irrelevant filler content to artificially reach target lengths, reducing the relevance and effectiveness of assessments. To address these limitations, we introduce NeedleBench, a comprehensive synthetic framework designed to assess retrieval and reasoning performance in bilingual long-context tasks with adaptive context lengths (e.g., 32k, 128k, and beyond). NeedleBench systematically embeds key data points at varying depths to rigorously test models' capabilities in diverse settings. Tasks within NeedleBench are categorized into two distinct scenarios: information-sparse, characterized by minimal relevant details embedded within extensive irrelevant text to simulate simpler real-world retrieval tasks; and information-dense, implemented as the Ancestral Trace Challenge, where relevant information is continuously distributed throughout the context to simulate more complex real-world reasoning tasks. Our experiments show that, while recent reasoning models such as Deepseek-R1 and OpenAI's o3 have demonstrated strong performance on mathematical reasoning benchmarks, they still struggle to generalize their reasoning abilities and perform poorly on our information-dense tasks, frequently encountering difficulties with continuous retrieval and reasoning even at relatively shorter context lengths. Furthermore, we identify and characterize a phenomenon termed 'under-thinking', wherein models prematurely conclude their reasoning processes despite the availability of relevant information. NeedleBench thus provides critical insights and targeted evaluation tools essential for understanding and improving the long-context capabilities of LLMs. All codes and resources are publicly available at OpenCompass.

---

[*]Co-corresponding authors.

# 1 Introduction

The capability of LLMs to process long texts is particularly crucial across various situations (Mohtashami & Jaggi, 2023; Grattafiori et al., 2024; Yang et al., 2025; Team et al., 2024; Tunstall et al., 2023a; Team, 2025; Yang et al., 2024b; DeepSeek-AI, 2025; Lyu et al., 2025). LLMs can rapidly identify and summarize relevant information within lengthy documents, making them invaluable for legal document retrieval, academic research, and aggregating business intelligence, among other applications (Wang et al., 2024; Lee et al., 2024). To meet these needs, modern close-sourced LLMs have recently been developed to support longer context windows (OpenAI, 2023; Anthropic, 2024b; Gemini Team, 2024; Cai et al., 2024; GLM et al., 2024; Bai et al., 2023a). As models accommodate longer text lengths, verifying their comprehension of details within the text becomes increasingly essential.

A variety of approaches have been proposed to evaluate the long-context capabilities of LLMs, though assessing performance at extremely long contexts (e.g., around 1M tokens) remains challenging. Early methods embed crucial "passkeys" in repetitively structured texts to test information retrieval over long sequences (Mohtashami & Jaggi, 2023; Zhang et al., 2023). Building on this idea, the Needle In A Haystack (NIAH) test (Kamradt, 2023) introduces more realistic settings by using non-repetitive personal essays as filler material, increasing task complexity and extending context lengths up to 200K tokens. According to the results reported in Kamradt (2023), advanced models such as Claude 2.1 and GPT-4 Turbo generally perform well on these targeted extraction tasks (Anthropic, 2024b).

More recent benchmarks like LongBench v2 (Bai et al., 2024; 2023b) offer diverse comprehension tasks but typically fix task length and lack adaptability, while Ruler (Hsieh et al., 2024) attempts adaptive-length evaluations by inserting irrelevant text. However, inserting too much irrelevant content can make it easy for models to complete tasks by focusing only on a few key points, without truly reading the full context. This may fail to reflect how models perform in more information-dense tasks that require careful reading and integration of content. For instance, in legal case analysis, an LLM must extract relevant facts and legal provisions from case files and synthesize them to answer specific questions. In such cases, **even small details in the long context can affect the final judgment**, making the input highly information-dense with little room for irrelevant content. Furthermore, benchmarks derived from real-world texts often encounter the issue of models leveraging prior knowledge acquired during pre-training, thereby undermining a true assessment of long-context understanding.

Given these limitations, it is crucial to design benchmarks that not only support adaptive context lengths, but also **feature information-dense tasks** where relevant content is distributed throughout the input. Such tasks should minimize reliance on irrelevant filler used in Kamradt (2023); Hsieh et al. (2024), ensuring that models must engage with the entire context to perform well. This helps better evaluate a model's true capacity for long-context understanding, while also avoiding scenarios where models can simply rely on memorized knowledge from pre-training instead of actually processing the input text (Bai et al., 2023b; 2024).

To address the limitations of existing long-context evaluation methods, we present *NeedleBench*, a dataset framework that encompasses both information-sparse and information-dense tasks. *NeedleBench* is designed to provide a comprehensive, targeted assessment of models' abilities to extract, analyze, and reason over long texts. In particular, our benchmark supports flexible context lengths (4k, 8k, 32k, 128k, 200k, 1000k, and beyond), allowing strategic insertion of key data points at various depths to rigorously test retrieval and reasoning skills. Moreover, these tasks are largely synthetic, which helps mitigate the influence of prior internal knowledge and compels models to truly process the given context.

Within *NeedleBench*, we include tasks that continue the tradition of inserting irrelevant filler (e.g., Single-Needle Retrieval, Multi-Needle Retrieval, Multi-Needle Reasoning), providing a information-sparse baseline for evaluating how well models handle straightforward retrieval in extended texts. On the other hand, we propose the Ancestral Trace Challenge (ATC), an information-dense task designed to reflect more complex real-world scenarios that require continuous logical reasoning. Our findings reveal that, despite recent top-performing models such as OpenAI's o3(OpenAI, 2025b) and DeepSeek-R1 (DeepSeek-AI, 2025) achieving impressive results on mathematical benchmarks such as AIME (Di Zhang, 2025) and MATH500 (Hendrycks et al., 2021),

they still struggle to **generalize their reasoning abilities** and perform poorly on our information-dense tasks. Our major contributions are as follows:

- **Comprehensive Bilingual Long-Context Benchmark:** We introduce *NeedleBench*, a customizable framework for evaluating bilingual long-context capabilities of LLMs across multiple length intervals, covering both information-sparse and information-dense tasks.

- **Long-Context Information-Dense Task:** We design the Ancestral Trace Challenge, simulating real-world information-dense tasks where models must track interdependent entities and constraints across evolving contexts. Our experiments demonstrate that current LLMs still struggle with complex long-context tasks.

- **Fine-Grained Evaluation and Analysis:** We offer an in-depth assessment of mainstream models' retrieval and reasoning performance under different context conditions. All reproducible scripts, code, and datasets will be made available upon publication.

## 2  Related Work

**Long-Context Language Models.**  Recent large language models have rapidly increased their context window sizes, from early 4K-token models like Longformer and BigBird (Beltagy et al., 2020; Zaheer et al., 2021) to commercial models such as GPT-4 Turbo (128K), Claude 3 (200K), and Gemini 1.5 Pro (1M) (OpenAI, 2023; Anthropic, 2024a; Gemini Team, 2024). While many models achieve near-perfect scores on NIAH test, these results do not guarantee strong reasoning or comprehension in information-dense settings. Recent work (Hsieh et al., 2024) shows that passing retrieval tests does not always mean robust understanding, underscoring the need for more challenging benchmarks to truly assess long-context reasoning.

**Long-Context Benchmarks.**  Existing long-context benchmarks present a trade-off: programmatically constructed tests like the Needle In A Haystack (NIAH) test (Kamradt, 2023), Ruler (Hsieh et al., 2024), and MRCR (Vodrahalli et al., 2024) mitigate data contamination but are often limited to information-sparse retrieval, while real-world text benchmarks like LongBench v2 (Bai et al., 2024) offer complex reasoning but risk evaluating memorization due to potential pre-training data overlap. *NeedleBench* addresses this dilemma by programmatically constructing tasks across a spectrum of information densities, from sparse retrieval to our novel, information-dense Ancestral Trace Challenge (ATC). *NeedleBench* enables a fair assessment of a model's intrinsic long-context multi-step reasoning capabilities, free from the confound of prior knowledge, and provides a timely benchmark to evaluate the latest generation of reasoning LLMs (OpenAI, 2025b; DeepSeek-AI, 2025; Anthropic, 2024a).

## 3  Tasks and Datasets

We categorize *NeedleBench* tasks into two types: **Information-Sparse Tasks**, where only a small fraction of the input text contains relevant information for answering the question; and **Information-Dense Tasks**, where every sentence contains essential content and the model must fully comprehend all details to succeed. The overall structure of these tasks are illustrated in Fig. 1.

### 3.1  *NeedleBench* Information-Sparse Tasks

The information-sparse tasks in *NeedleBench* is composed of the following three tasks, each targeting a specific capability of long-context processing:

- **Single-Needle Retrieval Task (S-RT):** Tests LLMs' ability to recall a **single key** information inserted at various positions in a long text, highlighting their precision in navigating and recalling single detail within extensive texts.

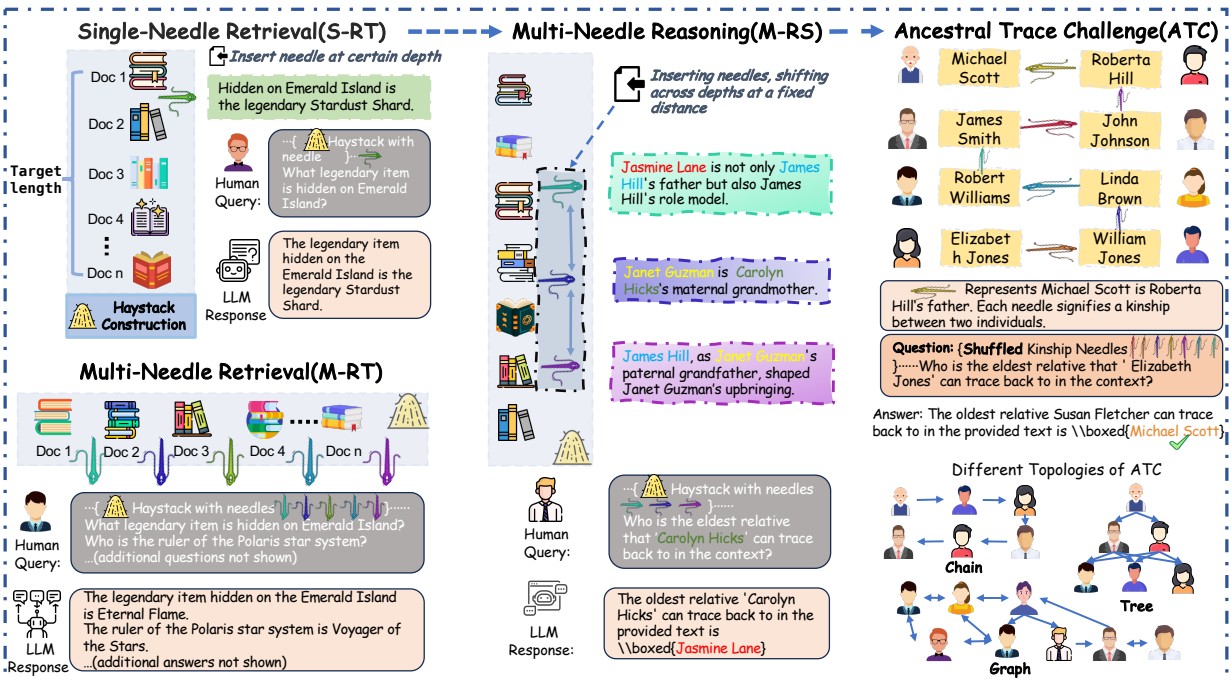

Figure 1: *NeedleBench* Framework. Our benchmark consists of two main categories: Information-Sparse Tasks (left two columns), which include Single-Needle Retrieval, Multi-Needle Retrieval, and Multi-Needle Reasoning with irrelevant filler content; and Information-Dense Tasks (rightmost column), specifically the ATC, designed to eliminate irrelevant filler and require comprehensive understanding of all content.

- **Multi-Needle Retrieval Task (M-RT):** Explores LLMs' ability to retrieve **multiple pieces** of related information scattered across a lengthy text, simulating complex real-world queries that require extracting several data points from comprehensive documents.

- **Multi-Needle Reasoning Task (M-RS):** Evaluates LLMs' ability for complex reasoning by extracting **multiple pieces** of information (ranging from 2 to 5 key facts) from long texts and using them to logically answer questions that demand an integrated understanding and reasoning of various text segments.

In *NeedleBench* tasks, both the "needles" (key information units) and the "haystack" (background or filler content) are carefully constructed to ensure a fair and challenging evaluation. The needles are **synthetic, abstract, and fictional** statements or relational facts, deliberately designed to avoid overlap with any real-world knowledge or pretraining data. We further discuss the necessity of using synthetic data for fair evaluation in Appendix D.

For retrieval tasks, these may be unique fabricated facts (e.g., "Hidden on Emerald Island is the legendary Stardust Shard"), while for reasoning tasks, they are synthetic kinship needles, which are the same as those used in the information-dense task; see Sec. 3.2 for details. The haystack for English tasks is built by extending the prompt with passages from the PaulGrahamEssays dataset (Kamradt, 2023), and for Chinese tasks, we use the ChineseDomainModelingEval dataset (Wei et al., 2023b) to ensure linguistic diversity and high-quality filler content.

**Evaluation Metrics.** To quantitatively evaluate model performance on the information-sparse tasks, we adopt a keyword-aware scoring approach that emphasizes the successful retrieval of core information from long texts. For each instance, a predefined set of core keywords is used to determine whether the model prediction sufficiently captures the essential content.

We employ a keyword-aware variant of the Exact Match (EM) metric in Eq. (1). For each test, a predefined set of core keywords $W_{c,d,r}$ is compared with the model's prediction $P_{c,d,r}$, where $c$ represents the context length, $d$ represents the needle depth or position, and $r$ represents the index of repetition. Full credit is awarded if any keyword from $W_{c,d,r}$ appears in $P_{c,d,r}$, and zero otherwise:

$$\text{Score}_{c,d,r} = \begin{cases} 100, & \text{if } P_{c,d,r} \cap W_{c,d,r} \neq \emptyset, \\ \\ 0, & \text{otherwise.} \end{cases} \tag{1}$$

For the Multi-Needle Reasoning task, the model's prediction $P_{c,d,r}$ is post-processed to extract only the content inside the `\boxed{...}` (i.e., $P_{c,d,r} = P_{c,d,r}^{\text{box}}$). For each task, we first compute a task-specific score by averaging across different experimental dimensions.

$$\text{Task Score} = \frac{1}{|C| \cdot |D| \cdot R} \sum_{c \in C} \sum_{d \in D} \sum_{r=1}^{R} \text{Score}_{c,d,r} \tag{2}$$

where $C$ represents the set of context lengths, $D$ represents the set of needle depths or positions, and $R$ is the number of repetitions for each configuration. The overall benchmark score is then calculated as a weighted average across all information-sparse tasks:

$$\text{Overall Score} = \frac{1}{|\mathcal{T}|} \sum_{t \in \mathcal{T}} \text{Task Score}_t, \quad \mathcal{T} = \{\text{S-RT}, \text{M-RT}, \text{M-RS}\} \tag{3}$$

By evaluating performance across different context lengths, needle positions, and task types, *NeedleBench* provides a detailed assessment of a model's long-context abilities. For each configuration, we repeat the test $R = 10$ times to enhance result stability. Token lengths are measured using the GPT-4 tokenizer[1].

To mitigate the risk of instruction truncation—where essential prompt instructions at the end of the context may be lost due to tokenizer discrepancies across different models—we subtract a buffer from the target context length when generating each input. This buffer ensures that, despite differences in how various tokenizers segment the same prompt, all models are consistently exposed to the complete instructions, thereby enabling a fair and reliable evaluation across models with different tokenizers.

### 3.2 *NeedleBench* Information-Dense Tasks

Unlike information-sparse tasks that often include large portions of irrelevant filler content, the Ancestral Trace Challenge (ATC) is explicitly designed to be information-dense: every sentence in the input context contains critical information directly related to the target question. There is no irrelevant text—each piece of content is essential and contributes to determining the correct answer. The detailed generation algorithm for ATC is provided in Appendix E. The ATC task introduces diversity along several axes to comprehensively evaluate long-context reasoning capabilities:

- **Name and Relationship Diversity:** Each critical piece of information in ATC is assigned a unique, randomized name, and the relationships among entities are highly diverse—including but not limited to parent-child, ancestor-descendant (across multiple generations), and dual-role relationships (e.g., one individual may simultaneously be a parent and a lifelong mentor). This combined diversity ensures that every question features unpredictable entities and relational structures, preventing memorization and encouraging genuine reasoning.

- **Task Diversity:** ATC includes multiple question types, which can be categorized as follows: (1) identifying the eldest ancestor; (2) tracing the $n$-th ancestor of a given individual; (3) tracing the $n$-th

---

[1] https://github.com/openai/tiktoken

descendant of a given individual; (4) calculating the relationship distance between two individuals. This variety ensures that models are evaluated on a broad spectrum of reasoning skills.

- **Logical Complexity and Context Length Diversity:** The number of needles per question is varied from 2 to 512, increasing both logical complexity and context length, requiring the model to continuously integrate information from multiple sources in the context.

**Evaluation Metrics.** For the information-dense ATC task, we use an exact match (EM) metric based on the model's ability to output the correct answer in the required format `\boxed{...}`, allowing for precise and automated evaluation. To aggregate results across different levels of task difficulty (i.e., different numbers of embedded needles), we compute a weighted average of the exact match accuracy, where the weight for each subtask is proportional to the number of needles it contains. This is similar to the weighted average metric in classification evaluation (Pedregosa et al., 2011), where each class is weighted by its support (number of instances). For each configuration, we repeat the evaluation $R = 10$ times to ensure stability.

Formally, let $P_{2^k}$ denote the exact-match accuracy (in percentage) of a model on the ATC subtask with $2^k$ needles, and let $\mathcal{N} = \{2^k \mid k = 1, 2, \ldots, 9\}$ be the set of needle counts. We report following two metrics:

$$\text{Weighted Average} = \frac{\sum_{N \in \mathcal{N}} (P_N \times N)}{\sum_{N \in \mathcal{N}} N} \qquad \text{ENL}_\tau = \max\left\{N \in \mathcal{N} \mid P_N \geq \tau\right\} \qquad (4)$$

where $\tau$ is a threshold (we use $\tau = 50\%$ and denote the metric as ENL-50). The ENL metric (Effective Needle Length) reflects the largest number of needles $N$ for which the model's exact-match accuracy $P_N$ remains at least $\tau$. In other words, ENL-50 measures the model's effective reasoning depth: the maximum task difficulty (needle count) at which the model can still achieve at least 50% accuracy.

## 4 Experiments

We evaluate mainstream open-source LLMs on the information-sparse tasks in *NeedleBench* at two representative context lengths: 32K and 128K tokens. Each model is tested at the maximum context length it officially supports. To enable direct comparison across model generations, we also include Qwen-2.5 models in the 32K setting, even though they support longer contexts. All evaluated models are instruction-tuned rather than base models. Results are shown in Tabs. 1 and 2. The full list of evaluated models and their context window sizes is provided in Appendix A.

We mainly discuss the performance of mainstream models without long chain-of-thought (long CoT) reasoning in information-sparse tasks in the main text. For models with long CoT/reasoning abilities, we focus on evaluating them on information-dense tasks. But we also provide their results on information-sparse tasks in Appendix B. For the information-dense ATC task, which is designed as a long-context evaluation with high information density and minimal irrelevant content, we expand our evaluation to include leading close-sourced API models such as GPT-4.1 (OpenAI, 2025a), OpenAI's o3-mini (OpenAI, 2025b), Claude-3.7-Sonnet-Thinking (Anthropic, 2024a), and DeepSeek R1 (DeepSeek-AI, 2025).

### 4.1 Performance of *NeedleBench* Information-Sparse Tasks

### 4.1.1 Impact of Model Architecture and Technical Advances on Retrieval Performance

In the 32K context setting, we observe a pronounced difference in retrieval performance not only between older and newer model generations, but also across different architectural and technical choices. Early-generation models such as Zephyr-7B-Beta (Tunstall et al., 2023b) and Qwen-1.5-1.8B (Bai et al., 2023a) achieve relatively low overall scores on the Single-Retrieval task, and often fail to achieve perfect recall—especially when the relevant information is located far from the end of the context, making it difficult for the model to maintain and utilize that information. See Fig. 2a and Fig. 2b for their performance.

This phenomenon can be partially attributed to the underlying architectural and technical designs of these models. Zephyr-7B-Beta, for example, employs sliding window attention (SWA) (Beltagy et al., 2020), which

Table 1: **Main Results of *NeedleBench* 32K.** Overall denotes the mean score across all tasks. **Bold** denotes the best score among all models, and underline denotes the best score under the same model scale. The same notation applies in the following tables.

| Model | Single-Retrieval | | | Multi-Retrieval | | | Multi-Reasoning | | | Overall |
|---|---|---|---|---|---|---|---|---|---|---|
| | Chinese | English | Overall | Chinese | English | Overall | Chinese | English | Overall | |
| *Models with Fewer Than 7B Parameters* | | | | | | | | | | |
| Qwen-2.5-1.5B | 96.67 | 94.44 | 95.56 | 95.39 | 97.29 | 96.34 | 0.00 | 15.63 | 7.82 | 66.57 |
| Qwen-1.5-4B | 95.66 | 99.60 | 97.63 | 95.76 | 97.01 | 96.38 | 2.68 | 7.05 | 4.86 | 66.29 |
| ChatGLM3-6B-32K | 93.64 | 98.89 | 96.26 | 90.83 | 94.38 | 92.61 | 0.18 | 9.07 | 4.62 | 64.50 |
| Qwen-1.5-1.8B | 78.99 | 71.11 | 75.05 | 54.26 | 52.93 | 53.60 | 0.00 | 0.00 | 0.00 | 42.88 |
| *Models with 7-20B Parameters* | | | | | | | | | | |
| Qwen-2.5-14B | 99.19 | 98.79 | 98.99 | **99.07** | 99.23 | 99.15 | 29.65 | 17.90 | 23.78 | 73.97 |
| Qwen-2.5-7B | **100.00** | 99.80 | 99.90 | 97.70 | 99.31 | 98.51 | 12.65 | 18.64 | 15.64 | 71.35 |
| Qwen-1.5-14B | 99.60 | 99.49 | 99.55 | 92.57 | 99.15 | 95.86 | 0.58 | 10.20 | 5.39 | 66.93 |
| Mistral-7B-Instruct-v0.2 | 92.73 | 96.36 | 94.55 | 87.23 | 96.97 | 92.10 | 11.57 | 14.27 | 12.92 | 66.52 |
| Zephyr-7B-Beta | 35.35 | 36.77 | 36.06 | 18.14 | 27.60 | 22.87 | 1.87 | 7.45 | 4.66 | 21.20 |
| *Models Larger Than 20B Parameters* | | | | | | | | | | |
| Qwen-2.5-72B | **100.00** | **100.00** | **100.00** | 98.71 | **99.96** | **99.33** | **39.80** | **52.07** | **45.93** | **81.76** |
| Qwen-2.5-32B | **100.00** | **100.00** | **100.00** | 98.71 | 95.72 | 97.21 | 33.31 | 38.96 | 36.14 | 77.78 |
| Qwen-1.5-32B | 99.60 | **100.00** | 99.80 | 98.02 | 98.95 | 98.48 | 11.67 | 14.85 | 13.26 | 70.51 |
| Mixtral-8x7B-Instruct-v0.1 | 95.76 | 99.60 | 97.68 | 94.63 | 99.43 | 97.03 | 5.93 | 15.88 | 10.91 | 68.54 |
| Qwen-1.5-72B | 97.37 | 89.60 | 93.48 | 93.49 | 92.24 | 92.87 | 9.75 | 7.35 | 8.55 | 64.97 |

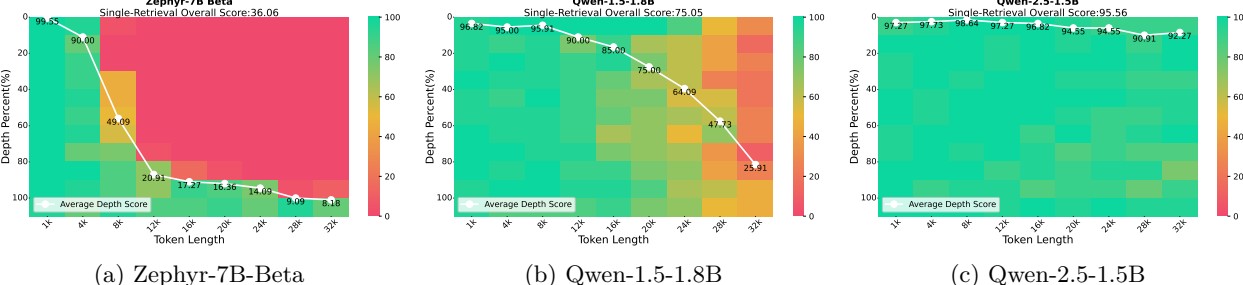

(a) Zephyr-7B-Beta     (b) Qwen-1.5-1.8B     (c) Qwen-2.5-1.5B

Figure 2: Comparison of different model generations on the Single-Retrieval Task. Newer models such as Qwen-2.5 show clear improvements over earlier models like Zephyr-7B-Beta and Qwen-1.5.

restricts attention to local windows and may hinder the propagation of information across distant tokens. Qwen-1.5 adopts a mixture of sliding window and full attention, which offers some improvement but still falls short of optimal long-range retrieval. In contrast, newer models such as Qwen-2.5 (Yang et al., 2024a) leverage advanced techniques for long-context extrapolation, including Dual Chunk Attention (DCA) (An et al., 2024) and YaRN (Peng et al., 2023), which enable effective modeling of both local and global dependencies. In practice, even at the small 1.5B scale, Qwen-2.5 achieves near-perfect or perfect scores on retrieval tasks (see Fig. 2c), indicating that these new architectural techniques can substantially enhance retrieval ability in long-context settings, and that accurate long-context retrieval is now commonly observed in modern LLMs.

It is also important to note that the use of local attention mechanisms such as SWA does not necessarily hinder strong long-context retrieval. For example, the recent Gemma-3 (Team et al., 2025) models employ a hybrid attention strategy, interleaving local and global attention layers at a 5:1 ratio (compared to the 1:1 ratio in Gemma-2). Despite having a higher proportion of local attention, Gemma-3 still achieves strong performance at very long context lengths, as demonstrated by the 128K context results (Tab. 2), suggesting carefully balancing the ratio of local and global attention layers is crucial for optimizing long-context performance.

### 4.1.2 Challenges in Multi-Needle Reasoning Compared to Retrieval Tasks

While retrieval tasks, such as Single-Needle Retrieval and Multi-Needle Retrieval, have become relatively straightforward for modern LLMs, the Multi-Needle Reasoning task presents a far greater challenge in Tab. 1. This task requires models to extract key information from multiple locations within a long context and

Table 2: **Main Results of *NeedleBench* 128K.** Most models achieve excellent performance on retrieval tasks, indicating that they can easily handle long-context retrieval. However, there is still significant room for improvement on Multi-Needle Reasoning tasks, with even the best model scoring below 50.

| Model | Single-Retrieval | | | Multi-Retrieval | | | Multi-Reasoning | | | Overall |
|---|---|---|---|---|---|---|---|---|---|---|
| | Chinese | English | Overall | Chinese | English | Overall | Chinese | English | Overall | |
| *Models with Fewer Than 10B Parameters* | | | | | | | | | | |
| InternLM3-8B | 99.09 | 99.66 | 99.38 | 96.00 | 98.91 | 97.45 | 23.44 | 35.85 | 29.64 | 75.49 |
| LLaMA-3.1-8B | **100.00** | **100.00** | **100.00** | 95.18 | 98.64 | 96.91 | 10.82 | 21.22 | 16.02 | 70.98 |
| Qwen-2.5-7B | 99.89 | 96.82 | 98.35 | 96.00 | 98.00 | 97.00 | 10.68 | 23.12 | 16.90 | 70.75 |
| GLM-4-9B-Chat | 98.98 | 88.41 | 93.69 | 97.32 | **99.91** | 98.61 | 4.40 | 10.06 | 7.23 | 66.51 |
| Gemma-3-4B | 95.23 | 89.89 | 92.56 | 83.00 | 86.77 | 84.89 | 15.28 | 16.34 | 15.81 | 64.42 |
| InternLM2.5-7B-Chat-1M | 99.43 | 99.66 | 99.55 | 90.95 | 98.55 | 94.75 | 14.57 | 11.88 | 13.22 | 69.17 |
| *Models with 10-20B Parameters* | | | | | | | | | | |
| Gemma-3-12B | 92.61 | 99.55 | 96.08 | 91.77 | 94.86 | 93.32 | 33.72 | 39.32 | 36.52 | 75.31 |
| Qwen-2.5-14B | 99.89 | 95.91 | 97.90 | 98.09 | 97.73 | 97.91 | 29.20 | 22.95 | 26.08 | 73.96 |
| *Models Larger Than 20B Parameters* | | | | | | | | | | |
| Gemma-3-27B | 96.70 | 98.98 | 97.84 | 94.18 | 96.36 | 95.27 | **47.93** | 48.15 | **48.04** | 80.38 |
| Qwen-2.5-32B | 99.43 | 99.77 | 99.60 | 98.91 | 99.68 | **99.30** | 32.19 | 39.55 | 35.87 | 78.25 |
| LLaMA-3.1-70B | **100.00** | 99.89 | 99.94 | **99.00** | 99.09 | 99.05 | 15.71 | 20.51 | 18.11 | 72.37 |
| Qwen-2.5-72B | 99.77 | **100.00** | 99.89 | 98.73 | 99.77 | 99.25 | 36.79 | **51.05** | 43.92 | **81.02** |

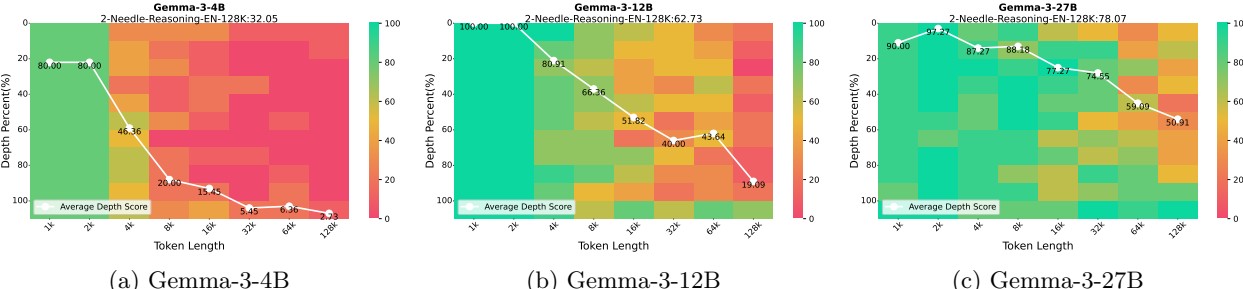

(a) Gemma-3-4B      (b) Gemma-3-12B      (c) Gemma-3-27B

Figure 3: Performance of Gemma-3 models with increasing parameter size on the 2-Needle Reasoning (EN, 128K) task. Larger models consistently achieve higher scores, highlighting the benefit of increased capacity for multi-point reasoning in long contexts.

integrate these interdependent pieces to perform complex reasoning. Only a few large-scale models—such as Qwen-2.5-72B and Qwen-2.5-32B—demonstrate significantly higher scores (with Qwen-2.5-72B achieving the best result at 45.93%, still below 50%), indicating their superior ability to handle such reasoning demands. In contrast, smaller models or earlier generations, particularly those below 20B parameters, struggle to perform effectively, often achieving near-zero scores. This highlights the inherent difficulty of reasoning tasks that demand multi-point integration over densely packed, interrelated information within long sequences.

When scaling to 128K context length, we find that retrieval remains a largely solved problem for modern LLMs. Most models that support this length are relatively recent, and as such, exhibit strong performance on both Single and Multi-Needle Retrieval tasks, as shown in Tab. 2. Similar to the trends observed in the 32K setting (Tab. 1), Multi-Needle Reasoning continues to reveal substantial performance gaps across models. Notably, InternLM3-8B stands out among sub-10B models, achieving reasoning performance comparable to mid-sized models like Qwen-2.5-14B, though it still trails behind the strongest models in the 20B+ range.

### 4.1.3 Effect of Model Scale on Multi-Needle Reasoning Performance

To provide a clear visualization of the impact of model parameter size on Multi-Needle Reasoning, we present the performance of the Gemma-3 series (4B, 12B, and 27B) on the 2-Needle Reasoning task at 128K context length. As shown in Fig. 3, as the model size increases, the color in the figure gradually shifts to green, indicating higher scores and stronger reasoning ability for larger models in integrating multiple information points within long contexts.

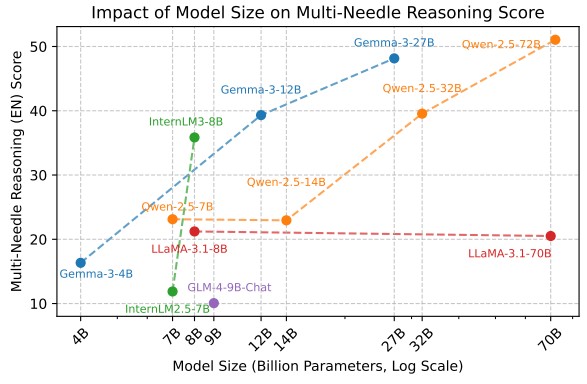
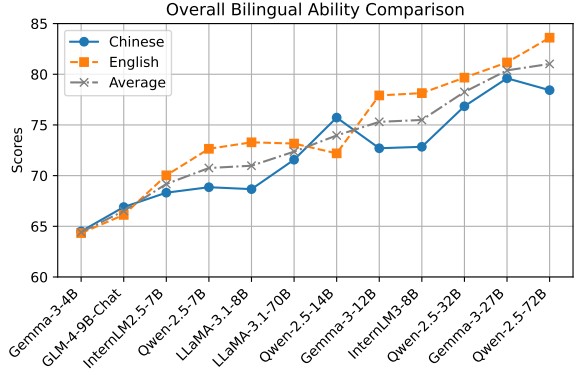

(a) Model size vs. long-context reasoning ability.

(b) English vs. Chinese performance.

Figure 4: Impact of language and model size on NeedleBench 128K performance. Left: Increasing model parameter size generally leads to better long-context reasoning. Right: Most models exhibit a clear performance gap between English and Chinese, with English scores typically higher.

Figure 4a demonstrates a clear positive correlation between model parameter size and Multi-Needle Reasoning (EN) performance, with a marked improvement observed beyond the 10B–20B parameter range, highlighting model scale as a key factor for long-context reasoning. Within individual model series, such as Gemma and Qwen, reasoning scores consistently increase with scale, reflecting classic scaling laws (Kaplan et al., 2020); however, this trend does not universally apply, as the LLaMA-3.1 series shows minimal performance gains from 8B to 70B parameters. While most small models (<10B) achieve low scores, certain models like InternLM3-8B outperform some larger counterparts, suggesting that training strategies, architecture, or fine-tuning can significantly boost small model capabilities. Notably, models with similar parameter counts from different series can exhibit substantial performance differences—for example, Gemma-12B-IT significantly outperforms Qwen-2.5-14B—indicating that architecture, pretraining data, and instruction tuning play crucial roles. Finally, the LLaMA series displays a saturation effect, with scores plateauing around 20 despite increasing parameters, implying that scaling alone is insufficient and further improvements require additional optimization strategies.

### 4.1.4 Effect of Needle Count on Multi-Needle Reasoning Performance

To further analyze how the number of reasoning points (needles) affects model performance, we visualize the results of Gemma-3-27B on the Multi-Needle Reasoning (ZH, 128K) task as the number of needles increases from 2 to 5, where "ZH" indicates that the task is conducted in Chinese. As shown in Fig. 5, the heatmaps become progressively redder, indicating a clear decline in performance as the reasoning complexity grows. This trend highlights the significant challenge that even strong models face when required to integrate a larger number of interdependent information points within long contexts. While retrieval over long context windows is becoming a solved problem for modern LLMs, integrating multiple critical details to form a coherent answer remains a significant challenge for all models. More advanced and larger-scale models generally perform better on such tasks, but even the strongest models are still far from perfect in this regard. We provide more detailed analysis on Multi-Needle Reasoning tasks in Appendix C.

### 4.1.5 Impact of Language: Which Model Performs Better under the Bilingual Scenario?

To directly address which model performs best in the bilingual scenario, we analyze the overall NeedleBench 128K performance for both English and Chinese, as shown in Fig. 4b. **Qwen2.5-72B** achieves the highest scores in both languages, indicating the strongest bilingual performance. Across most evaluated models, there is generally a performance gap between English and Chinese, with English results typically being higher. Performance in English tends to be higher than in Chinese for most models, though this is not universal; for example, Qwen2.5-14B and GLM4-9B achieve slightly better results in Chinese. Such differences may

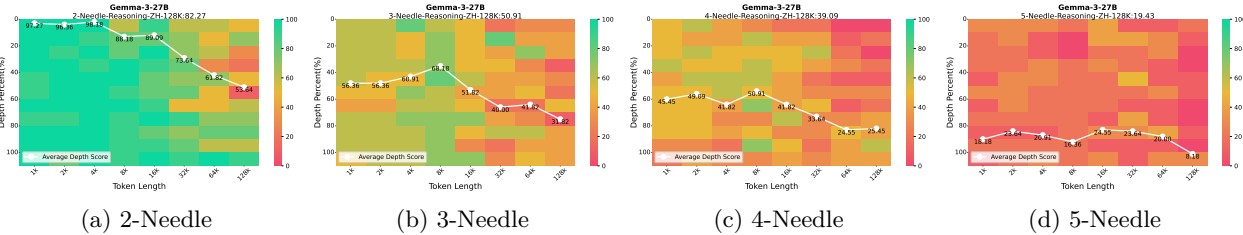

|              |              |              |              |
|:------------:|:------------:|:------------:|:------------:|
| (a) 2-Needle | (b) 3-Needle | (c) 4-Needle | (d) 5-Needle |

Figure 5: Performance of Gemma-3-27B on Multi-Needle Reasoning as the number of needles increases. The color shift towards red in the heatmaps indicates a clear decline in performance as the task complexity grows, highlighting the increasing challenge of integrating more information points within long contexts.

Table 3: ATC task results for all evaluated models. Models with stronger reasoning abilities generally achieve higher scores, with DeepSeek-R1 achieving the best overall performance (total score 44.01). In contrast, models with fewer parameters often achieve only single-digit scores.

| Model | Needle Count | | | | | | | | | Evaluation Metric | | |
|---|---|---|---|---|---|---|---|---|---|---|---|---|
| | **2** | **4** | **8** | **16** | **32** | **64** | **128** | **256** | **512** | | | |
| | | | | Context Length (tokens) | | | | | | Weighted Score | | ENL |
| | **0.4K** | **0.5K** | **0.6K** | **0.7K** | **1.0K** | **1.5K** | **2.7K** | **5K** | **9.6K** | **≤ 2K** | **All** | **ENL-50** |
| *Closed-Source and Reasoning Models* | | | | | | | | | | | | |
| Claude-3.7-Sonnet-Thinking | **100.0** | **100.0** | 92.5 | 92.5 | 67.5 | 40.0 | 15.0 | 7.5 | 2.5 | 59.84 | 12.39 | 32 |
| DeepSeek-R1 | **100.0** | **100.0** | 87.5 | 95.0 | **97.5** | **90.0** | 70.0 | 65.0 | **15.0** | **92.86** | **44.01** | **256** |
| GPT-4o | **100.0** | 72.5 | 82.5 | 42.5 | 17.5 | 0.0 | 0.0 | 0.0 | 0.0 | 18.97 | 2.34 | 8 |
| GPT-4.1 | **100.0** | 95.0 | 87.5 | 82.5 | 75.0 | 62.5 | 37.5 | 2.5 | 0.0 | 71.43 | 14.13 | 64 |
| o3-mini | 97.5 | **100.0** | 97.5 | 92.5 | 82.5 | 30.0 | 0.0 | 0.0 | 0.0 | 58.85 | 7.26 | 32 |
| QwQ-32B | **100.0** | 97.5 | 92.5 | 65.0 | 32.5 | 12.5 | 0.0 | 0.0 | 0.0 | 33.41 | 4.12 | 16 |
| OREAL-32B | 92.5 | 55.0 | 45.0 | 20.0 | 22.5 | 7.5 | 5.0 | 0.0 | 0.0 | 18.13 | 2.86 | 4 |
| DeepSeek-R1-Qwen-32B | 95.0 | 75.0 | 47.5 | 35.0 | 12.5 | 5.0 | 0.0 | 0.0 | 0.0 | 17.06 | 2.10 | 4 |
| DeepSeek-R1-Qwen-14B | **100.0** | 62.5 | 37.5 | 22.5 | 5.0 | 0.0 | 0.0 | 0.0 | 0.0 | 10.08 | 1.24 | 4 |
| DeepSeek-R1-Qwen-7B | 77.5 | 25.0 | 2.5 | 0.0 | 0.0 | 0.0 | 0.0 | 0.0 | 0.0 | 2.18 | 0.27 | 2 |
| *Models with 20B or More Parameters* | | | | | | | | | | | | |
| Qwen1.5-72B | 70.0 | 25.0 | 10.0 | 0.0 | 0.0 | 0.0 | 0.0 | 0.0 | 0.0 | 2.54 | 0.31 | 2 |
| Qwen2.5-72B | 92.5 | 62.5 | 45.0 | 10.0 | 0.0 | 0.0 | 2.5 | 0.0 | 0.0 | 7.58 | 1.25 | 4 |
| Qwen1.5-32B | 75.0 | 20.0 | 7.5 | 7.5 | 5.0 | 0.0 | 0.0 | 0.0 | 0.0 | 4.52 | 0.56 | 2 |
| Qwen2.5-32B | 97.5 | 62.5 | 27.5 | 17.5 | 5.0 | 2.5 | 0.0 | 0.0 | 0.0 | 10.04 | 1.24 | 4 |
| Gemma-3-27B | 82.5 | 70.0 | 67.5 | 47.5 | 30.0 | 2.5 | 5.0 | 0.0 | 0.0 | 22.74 | 3.43 | 8 |
| *Models with 7-20B Parameters* | | | | | | | | | | | | |
| Qwen1.5-14B | 52.5 | 17.5 | 10.0 | 2.5 | 2.5 | 0.0 | 0.0 | 0.0 | 0.0 | 2.98 | 0.37 | 2 |
| Qwen2.5-14B | 72.5 | 47.5 | 25.0 | 7.5 | 5.0 | 0.0 | 0.0 | 0.0 | 0.0 | 6.47 | 0.80 | 2 |
| Gemma-3-12B | 72.5 | 55.0 | 45.0 | 17.5 | 10.0 | 2.5 | 0.0 | 0.0 | 0.0 | 11.79 | 1.45 | 4 |
| Mixtral-8x7B | 50.0 | 12.5 | 15.0 | 7.5 | 0.0 | 0.0 | 0.0 | 0.0 | 0.0 | 3.10 | 0.38 | 2 |
| GLM-4-9B | 52.5 | 35.0 | 10.0 | 7.5 | 2.5 | 0.0 | 0.0 | 0.0 | 0.0 | 4.17 | 0.51 | 2 |
| InternLM3-8B | 55.0 | 32.5 | 27.5 | 20.0 | 2.5 | 0.0 | 0.0 | 5.0 | 0.0 | 6.83 | 2.09 | 2 |
| *Models with Fewer Than 7B Parameters* | | | | | | | | | | | | |
| Qwen1.5-1.8B | 5.0 | 0.0 | 0.0 | 0.0 | 0.0 | 0.0 | 0.0 | 0.0 | 0.0 | 0.08 | 0.01 | 0 |
| Qwen1.5-4B | 35.0 | 10.0 | 7.5 | 0.0 | 0.0 | 0.0 | 0.0 | 0.0 | 0.0 | 1.35 | 0.17 | 0 |
| Qwen2.5-1.5B | 37.5 | 25.0 | 2.5 | 0.0 | 0.0 | 0.0 | 0.0 | 0.0 | 0.0 | 1.55 | 0.19 | 0 |
| Qwen2.5-7B | 75.0 | 37.5 | 7.5 | 5.0 | 5.0 | 0.0 | 0.0 | 0.0 | 0.0 | 4.76 | 0.59 | 2 |
| Mistral-7B | 40.0 | 17.5 | 2.5 | 2.5 | 0.0 | 0.0 | 0.0 | 0.0 | 0.0 | 1.67 | 0.21 | 0 |
| Gemma-3-4B | 57.5 | 27.5 | 15.0 | 17.5 | 2.5 | 0.0 | 0.0 | 0.0 | 0.0 | 5.60 | 0.69 | 2 |
| ChatGLM3-6B-32K | 27.5 | 7.5 | 10.0 | 5.0 | 2.5 | 0.0 | 2.5 | 0.0 | 2.5 | 2.58 | 1.88 | 0 |
| InternLM2.5-7B-1M | 60.0 | 27.5 | 15.0 | 2.5 | 5.0 | 0.0 | 0.0 | 0.0 | 0.0 | 4.37 | 0.54 | 2 |

be related to factors including pretraining data distribution, tokenization strategies, or language-specific modeling approaches. These observations suggest that further research is needed to improve cross-lingual generalization and to develop more robust multilingual long-context models.

### 4.2 *NeedleBench* Information-Dense Task

We present the results of the ATC task in Tab. 3, which evaluates model performance under different context lengths determined by the number of embedded factual units ('needles'). Here, "reasoning models" in Tab. 3 refer to models that explicitly output a "think" step or intermediate reasoning process before giving the final answer. As the needle count increases, the input context becomes longer and the question increasingly requires the model to perform continuous retrieval and reasoning.

Across all models, we observe a clear downward trend in performance as the number of needles increases in Fig. 6. Most models fail entirely beyond the 64-needle level, indicating that longer, information-dense contexts remain a major challenge. Additionally, **model size** plays a significant role: larger models tend to achieve higher scores. For example, within the Gemma family, performance steadily improves from 5.60 (4B) to 22.74 (27B), demonstrating the benefit of increased capacity in tackling information-dense tasks. When focusing on the average performance under short contexts ($\leq$2K tokens), the Gemma-3 series shows strong results across all model scales. In fact, Gemma-3 models of sizes 4B, 12B, and 27B each achieve the best score in their respective size categories, highlighting the series' robustness in low-to-medium complexity settings.

**Can State-of-the-Art Reasoning Models Generalize to Long-Chain Reasoning?** We use the ENL-50 metric to quantify the effective reasoning depth of each model—that is, the maximum number of compositional steps a model can reliably handle. As shown in Tab. 3, small models ($\leq$7B) can only generalize to 2-step reasoning, medium models (7–20B) up to 4 steps, and large models (>20B) up to 8 steps, with only the strongest models such as GPT-4.1 (ENL-50 = 64) and DeepSeek R1 (ENL-50 = 256) demonstrating the ability to generalize to much longer reasoning chains. Notably, the DeepSeek-R1-Qwen distillation models, which are trained by distilling DeepSeek R1's reasoning data, still perform quite poorly on our synthetic tasks: for example, DeepSeek-R1-Qwen-32B achieves an ENL-50 of only 4, and the 14B and 7B variants also fail to generalize beyond 2–4 steps. This indicates that, although these models may have memorized reasoning patterns from their teacher, they struggle to transfer such reasoning to broader, more diverse compositional tasks, highlighting the challenge of achieving true generalizable reasoning beyond rote pattern replication.

**The "Under-Thinking" Bottleneck in Information-Dense Long-Context Tasks.** To investigate why state-of-the-art (SOTA) models like o3-mini and DeepSeek R1 frequently fail at information-dense tasks, we manually annotated approximately 10% of the observed errors from the ATC task, focusing specifically on cases where the question type is "identifying the eldest ancestor." The main error types identified in our analysis are summarized in Table 4. For each error category, we provide representative prompt and response examples in Appendices H.1 to H.5 to illustrate how these errors occur in practice.

Our analysis reveals that the most prevalent failure mode among strong models is what we term *under-thinking*: models prematurely conclude that no further inference can be made, even when clear clues remain in the context. While the term "under-thinking" has previously been used to describe models abandoning a reasoning path too early in favor of another (often in math problem solving) (Wang et al., 2025), here we use it to refer to a distinct phenomenon: the inability to sustain inference by fully leveraging all relevant information, resulting in reasoning that **halts midway under the mistaken belief that nothing more can be inferred.** This under-thinking bottleneck, together with other characteristic errors, highlights the persistent challenges faced by LLMs in reliably handling information-dense, long-context tasks.

Beyond under-thinking errors, our analysis reveals several other characteristic failure modes in current LLMs on the ATC task. Instruction following errors are predominantly observed in smaller-parameter models, which often fail to comply with required output formats. Detailed statistics on instruction following errors are provided in Appendix F. Partial understanding errors occur when a model fixates on part of a statement—such as interpreting "Dan Newton is more than just a mother; Dan Newton is a lifelong mentor of Andrew Williams" as indicating only a mentorship—while overlooking the explicit familial role, in this case, "mother," thus breaking the full kinship chain. Repetitive output errors are frequently seen in models such as Deepseek-R1-Distill-Qwen-7B, suggesting a potential drawback of directly distilling large model outputs into smaller models via supervised fine-tuning. In addition, we observe other errors, such as misunderstanding the question intent, misinterpreting examples in the prompt as information to be used in reasoning, or making logical mistakes in the reasoning process. These issues are more prevalent in models with generally weaker

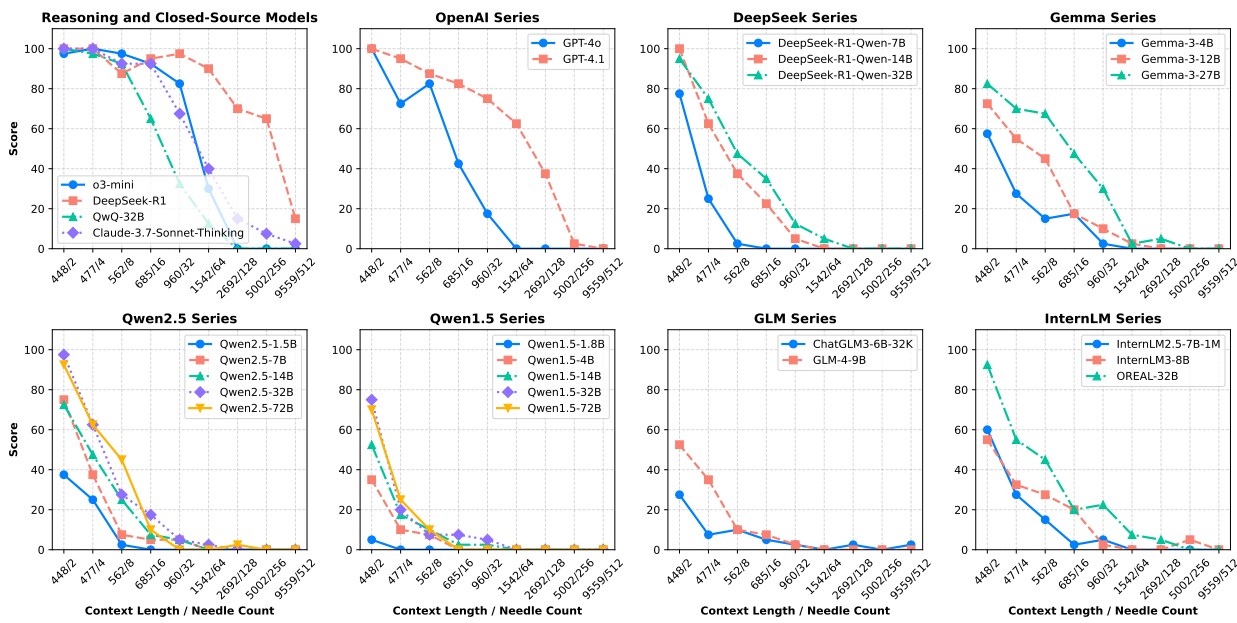

Figure 6: Performance decline trend of various models on ATC. OpenAI models (GPT-4 series) are shown separately for clarity. Notably, DeepSeek-R1 and GPT-4.1 exhibit a much slower decline in performance as the number of needles increases, demonstrating strong resistance to performance degradation on information-dense tasks. In contrast, most other models experience a rapid drop to near-zero scores as the length of the information-dense context increases.

Table 4: Analysis of Common Error Types in ATC Task: Under-thinking is the most prevalent error, especially in strong models like DeepSeek R1 and o3-mini. Partial understanding errors indicate models only grasp part of key information. Smaller models frequently exhibit instruction following and repetitive output errors.

| Error Type (%) | Explanation | Representative Models (non-exhaustive) |
|---|---|---|
| Under-thinking (60.2%) | The model prematurely halts reasoning, asserting that no further inference can be made despite the presence of remaining clues. | Reasoning models (e.g., DeepSeek R1, GPT-4.1, o3-mini, Claude-3-7-Sonnet, InternLM3-8B-Instruct, Qwen2.5-1.5B-Instruct) |
| Partial Understanding Error (12.9%) | Model only identifies part of the relationships mentioned. | Gemma3 27B, InternLM3-8B-Instruct, Qwen2.5-7B-Instruct |
| Instruction Following Error (7.5%) | Model fails to follow the required output format. | Qwen1.5-1.8B-Chat, Qwen2.5-1.5B-Instruct |
| Repetitive Output (7.5%) | Model either repeats reasoning steps or outputs meaningless text. | Deepseek-R1-Distill-Qwen-7B, Qwen2.5-1.5B-Instruct |
| Hallucination Error (7.5%) | Model introduces information that is not present in the original text. | DeepseekR1, Qwen1.5-1.8B-Chat, Deepseek-R1-Distill-Qwen-7B, |
| Other Errors (4.5%) | Miscellaneous errors not covered by the above categories. | Various models |

overall performance. Collectively, these diverse error types highlight the persistent challenges faced by LLMs in reliably handling information-dense long-context tasks.

## 5 Conclusion and Future Work

In this research, we conduct a comprehensive evaluation of large language models (LLMs) on retrieval and reasoning tasks in long-context scenarios. Our results reveal that even state-of-the-art LLMs—including Claude 3.7 Sonnet-Thinking, o3-mini, and DeepSeek R1—exhibit notable shortcomings, especially on the Ancestral Trace Challenge, which is designed to test information-dense, multi-step retrieval and reasoning across extended texts. While recent long-context models have made progress in information retrieval, we find that they still struggle considerably when required to perform sustained, multi-step retrieval and reasoning over contexts where critical information is densely interwoven throughout the input.

Our research highlights the importance of targeted assessments in pinpointing and addressing critical gaps in LLMs' abilities to manage information-dense scenarios. The "under-thinking" phenomenon identified in our study further underscores the need to improve models' reasoning strategies—especially their tendency to prematurely conclude tasks even when additional evidence is available. Future work can include exploring reinforcement learning to help models improve their reasoning and mitigate under-thinking, as well as expanding NeedleBench to cover more diverse and realistic information-dense scenarios, since NeedleBench is a synthetic benchmark and may not fully reflect the complexity of real-world tasks.

### Acknowledgments

We thank Zhiwei Fei, Fengzhe Zhou, Hongwei Liu, Maosong Cao, Linchen Xiao, Zihan Ma, and Dongsheng Zhu for the valuable discussion. This work was supported by National Key R&D Program of China 2022ZD0161600, and Shanghai Oriental Talents Project BJZH2024070.

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

## A    Evaluated Models

The following Tab. 5 presents a list of models evaluated in this study, along with their maximum context lengths.

Table 5: Evaluated Models. We used LMDeploy (Contributors, 2023) and vLLM (Kwon et al., 2023) to accelerate the inference process. Unless otherwise specified, we use greedy decoding with temperature set to 0 for all model outputs.

| Series | Models | Context Window |
|---|---|---|
| Qwen | Qwen-1.5-1.8B, Qwen-1.5-4B, Qwen-1.5-14B, Qwen-1.5-32B, Qwen-1.5-72B Qwen-2.5-1.5B, Qwen-2.5-7B, Qwen-2.5-14B, Qwen-2.5-32B, Qwen-2.5-72B, QwQ-32B | 32K-128K |
| Zhipu AI | ChatGLM3-6B-32K, GLM-4-9B-Chat, GLM-4-9B-Chat-1M | 32K, 128K, 1M |
| InternLM | InternLM3-8B, InternLM2.5-7B-Chat-1M, OREAL-32B | 32K–1M |
| LLaMA | LLaMA-3.1-8B, LLaMA-3.1-70B | 128K |
| Mistral | Mistral-7B-Instruct-v0.2, Mixtral-8x7B-Instruct-v0.1 | 32K |
| Zephyr | Zephyr-7B-Beta | 32K |
| Gemma | Gemma-3-4B-IT, Gemma-3-12B-IT, Gemma-3-27B-IT | 128K |
| DeepSeek | DeepSeek-R1, DeepSeek-R1-Distill-Qwen-7B, DeepSeek-R1-Distill-Qwen-14B, DeepSeek-R1-Distill-Qwen-32B | 128K |
| OpenAI | GPT-4o, GPT-4.1, o3-mini, o4-mini | 128K–1M |
| Claude | Claude-3.7-Sonnet-Thinking | 200K |

## B    Performance of Long CoT Model on Information-Sparse Tasks at *NeedleBench* 128K

In this section, we provide the performance of Long Chain of Thought(CoT) models (Wei et al., 2023a) on information-sparse tasks at *NeedleBench* 128K. As shown in Tab. 6, models equipped with Long CoT capabilities generally achieve stronger results on reasoning tasks. For example, DeepSeek-R1 and its Distilled variants demonstrate clear improvements in multi-needle reasoning compared to standard models. However, we also observe that the DeepSeek-R1-Distill-Qwen-7B model makes errors on the Single-Needle Retrieval task. This may indicate that after Long CoT fine-tuning—likely focused on mathematical or reasoning tasks—its overall performance on long-context retrieval is not as strong as its original, non-fine-tuned version. Such fine-tuning may not specifically optimize for long-context retrieval, which could explain the observed issues on the 128K Single-Needle Retrieval task.

In Sec. 4.2, DeepSeek-R1 demonstrates extremely strong long-chain reasoning ability, being able to extrapolate up to 256 reasoning steps in ATC task. In the Multi-Needle-Reasoning Task, which involves at most five reasoning steps—i.e., integrating up to five key information points distributed throughout the long context—even the best models still face significant challenges when key information is sparsely distributed. Although models like DeepSeek-R1 can perform up to 256 reasoning steps in information-dense settings, they still struggle to reliably integrate scattered evidence in information-sparse long-context tasks. This suggests that current models are not yet able to stably perform complex reasoning over multiple dispersed information points in long-context scenarios.

## C    Detailed Multi-Needle Reasoning Performance at 32K and 128K

In this section, we present the detailed performance breakdown for the Multi-Needle Reasoning task at both 32K and 128K context lengths. The results are organized by the number of reasoning steps required: 2-needle, 3-needle, 4-needle, and 5-needle reasoning scenarios. As shown in Tab. 7 and Tab. 8, the performance generally degrades as the number of reasoning steps increases, demonstrating the challenge of multi-step reasoning

Table 6: **Results of *NeedleBench* 128K (with Long CoT Models).** We include several Long CoT reasoning models such as DeepSeek-R1, DeepSeek-R1-Distill-Qwen-7B/14B/32B, and o4-mini in this evaluation. These models generally achieve stronger performance on reasoning tasks. For example, DeepSeek-R1 achieves the highest Multi-Needle Reasoning score (74.13) among all models at 128K context length.

| Model | Single-Retrieval | | | Multi-Retrieval | | | Multi-Reasoning | | | Overall |
|---|---|---|---|---|---|---|---|---|---|---|
| | Chinese | English | Overall | Chinese | English | Overall | Chinese | English | Overall | |
| *Models with Fewer Than 10B Parameters* | | | | | | | | | | |
| InternLM3-8B | 99.09 | 99.66 | 99.38 | 96.00 | 98.91 | 97.45 | 23.44 | 35.85 | 29.64 | 75.49 |
| LLaMA-3.1-8B | **100.00** | **100.00** | **100.00** | 95.18 | 98.64 | 96.91 | 10.82 | 21.22 | 16.02 | 70.98 |
| Qwen-2.5-7B | 99.89 | 96.82 | 98.35 | 96.00 | 98.00 | 97.00 | 10.68 | 23.12 | 16.90 | 70.75 |
| GLM-4-9B-Chat | 98.98 | 88.41 | 93.69 | 97.32 | **99.91** | 98.61 | 4.40 | 10.06 | 7.23 | 66.51 |
| DeepSeek-R1-Distill-Qwen-7B | 41.95 | 46.91 | 44.43 | 41.95 | 46.91 | 44.43 | 10.00 | 16.65 | 13.32 | 34.06 |
| Gemma-3-4B | 95.23 | 89.89 | 92.56 | 83.00 | 86.77 | 84.89 | 15.28 | 16.34 | 15.81 | 64.42 |
| InternLM2.5-7B-Chat-1M | 99.43 | 99.66 | 99.55 | 90.95 | 98.55 | 94.75 | 14.57 | 11.88 | 13.22 | 69.17 |
| *Models with 10-20B Parameters* | | | | | | | | | | |
| Gemma-3-12B | 92.61 | 99.55 | 96.08 | 91.77 | 94.86 | 93.32 | 33.72 | 39.32 | 36.52 | 75.31 |
| Qwen-2.5-14B | 99.89 | 95.91 | 97.90 | 98.09 | 97.73 | 97.91 | 29.20 | 22.95 | 26.08 | 73.96 |
| DeepSeek-R1-Distill-Qwen-14B | 94.68 | 95.95 | 95.32 | 94.68 | 95.95 | 95.32 | 25.99 | 39.29 | 32.64 | 74.43 |
| *Models Larger Than 20B Parameters* | | | | | | | | | | |
| Gemma-3-27B | 96.70 | 98.98 | 97.84 | 94.18 | 96.36 | 95.27 | 47.93 | 48.15 | 48.04 | 80.38 |
| Qwen-2.5-32B | 99.43 | 99.77 | 99.60 | 98.91 | 99.68 | 99.30 | 32.19 | 39.55 | 35.87 | 78.25 |
| DeepSeek-R1-Distill-Qwen-32B | 97.18 | 98.14 | 97.66 | 97.18 | 98.14 | 97.66 | 44.57 | 46.45 | 45.51 | 80.28 |
| OREAL-32B | 96.64 | 96.82 | 96.73 | 96.64 | 96.82 | 96.73 | 31.70 | 47.10 | 39.40 | 77.62 |
| QwQ-32B | 98.50 | 98.05 | 98.27 | 98.50 | 98.05 | 98.27 | 61.16 | 63.52 | 62.34 | 86.30 |
| LLaMA-3.1-70B | **100.00** | 99.89 | **99.94** | 99.00 | 99.09 | 99.05 | 15.71 | 20.51 | 18.11 | 72.37 |
| Qwen-2.5-72B | 99.77 | **100.00** | 99.89 | 98.73 | 99.77 | 99.25 | 36.79 | 51.05 | 43.92 | 81.02 |
| o4-mini | 99.18 | 99.14 | 99.16 | 99.18 | 99.14 | 99.16 | 55.06 | 61.31 | 58.18 | 85.50 |
| DeepSeek-R1 | 99.32 | 99.91 | 99.61 | **99.32** | **99.91** | **99.61** | **70.03** | **78.24** | **74.13** | **91.12** |

Table 7: **Multi-Needle Reasoning Sub-dataset Results of NeedleBench-32K.** Qwen-2.5-72B achieves the best overall performance (45.93%), followed by Qwen-2.5-32B (36.14%). Performance consistently degrades as the number of needles increases across all models, with larger models generally outperforming smaller ones.

| Model | 2-Needle | | 3-Needle | | 4-Needle | | 5-Needle | | Overall |
|---|---|---|---|---|---|---|---|---|---|
| | Chinese | English | Chinese | English | Chinese | English | Chinese | English | |
| *Models with Fewer Than 7B Parameters* | | | | | | | | | |
| Qwen-1.5-1.8B | 0.00 | 0.00 | 0.00 | 0.00 | 0.00 | 0.00 | 0.00 | 0.00 | 0.00 |
| Qwen-2.5-1.5B | 0.00 | 25.45 | 0.00 | 16.97 | 0.00 | 14.04 | 0.00 | 6.06 | 7.82 |
| Qwen-1.5-4B | 4.75 | 9.19 | 1.82 | 6.87 | 3.54 | 4.34 | 0.61 | 7.78 | 4.86 |
| ChatGLM3-6B-32K | 0.61 | 10.51 | 0.00 | 8.38 | 0.10 | 7.68 | 0.00 | 9.70 | 4.62 |
| *Models with 7-20B Parameters* | | | | | | | | | |
| Qwen-2.5-7B | 30.81 | 33.03 | 10.00 | 20.20 | 6.16 | 8.38 | 3.64 | 12.93 | 15.64 |
| LLaMA-3.1-8B | 45.76 | 44.14 | 19.09 | 13.94 | 18.28 | 13.64 | **11.21** | 11.62 | 22.21 |
| Mistral-7B-Instruct-v0.2 | 20.81 | 25.76 | 16.97 | 19.90 | 6.36 | 5.25 | 2.12 | 6.16 | 12.92 |
| Qwen-1.5-14B | 1.92 | 21.62 | 0.20 | 11.82 | 0.00 | 0.51 | 0.20 | 6.87 | 5.39 |
| Qwen-2.5-14B | 79.60 | 32.53 | 16.16 | 14.34 | 18.28 | 15.15 | 4.55 | 9.60 | 23.78 |
| Zephyr-7B-Beta | 3.64 | 13.23 | 2.02 | 8.59 | 1.21 | 4.44 | 0.61 | 3.54 | 4.66 |
| *Models Larger Than 20B Parameters* | | | | | | | | | |
| Qwen-1.5-32B | 30.00 | 26.16 | 7.58 | 12.53 | 5.45 | 6.26 | 3.64 | 14.44 | 13.26 |
| Qwen-2.5-32B | 90.00 | 74.95 | 21.52 | 30.71 | 17.47 | 27.07 | 4.24 | 23.13 | 36.14 |
| Qwen-1.5-72B | 16.46 | 13.23 | 11.62 | 8.18 | 5.45 | 3.23 | 5.45 | 4.75 | 8.55 |
| Qwen-2.5-72B | **91.31** | **75.96** | **38.79** | **53.13** | **20.20** | **49.09** | 8.89 | **30.10** | **45.93** |
| Mixtral-8x7B-Instruct-v0.1 | 15.35 | 34.14 | 4.85 | 12.22 | 1.52 | 10.91 | 2.02 | 6.26 | 10.91 |
| LLaMA-3.1-70B | 54.14 | 41.82 | 19.49 | 23.33 | 6.67 | 9.09 | 7.58 | 9.80 | 21.49 |

over long contexts. The 32K results in Tab. 7 show that even at shorter context lengths, models struggle with complex multi-needle reasoning tasks. The 128K results in Tab. 8 further reveal that extending context length does not necessarily improve multi-step reasoning performance, suggesting that current models face fundamental limitations in integrating information across multiple scattered locations in long contexts.

Table 8: **Multi-Needle Reasoning Sub-dataset Results of NeedleBench-128K.** DeepSeek-R1 demonstrates superior performance (74.13%), followed by QwQ-32B (62.34%) and o4-mini (58.18%). Models with Long-CoT capabilities generally achieve better results.

| Model | 2-Needle | | 3-Needle | | 4-Needle | | 5-Needle | | Overall |
|---|---|---|---|---|---|---|---|---|---|
| | Chinese | English | Chinese | English | Chinese | English | Chinese | English | |
| *Models with Fewer Than 7B Parameters* | | | | | | | | | |
| InternLM3-8B | 44.89 | 61.48 | 26.48 | 37.95 | 14.20 | 24.89 | 8.18 | 19.09 | 29.64 |
| LLaMA-3.1-8B | 26.93 | 34.77 | 6.36 | 17.39 | 5.45 | 15.23 | 4.55 | 17.50 | 16.02 |
| Qwen-2.5-7B | 25.68 | 44.43 | 7.61 | 26.59 | 6.02 | 11.02 | 3.41 | 10.45 | 16.90 |
| GLM-4-9B-Chat | 4.66 | 17.73 | 4.09 | 3.98 | 6.02 | 7.73 | 2.84 | 10.80 | 7.23 |
| DeepSeek-R1-Distill-Qwen-7B | 20.34 | 31.25 | 8.52 | 13.52 | 6.93 | 12.05 | 4.20 | 9.77 | 13.32 |
| Gemma-3-4B | 30.68 | 32.05 | 13.30 | 14.20 | 11.02 | 9.55 | 6.14 | 9.55 | 15.81 |
| InternLM2.5-7B-Chat-1M | 34.66 | 15.23 | 10.80 | 9.32 | 6.93 | 13.41 | 5.91 | 9.55 | 13.22 |
| *Models with 7-20B Parameters* | | | | | | | | | |
| Gemma-3-12B | 62.50 | 62.73 | 28.30 | 41.48 | 29.55 | 25.68 | 14.55 | 27.39 | 36.52 |
| Qwen-2.5-14B | 75.91 | 44.32 | 18.98 | 20.23 | 15.11 | 18.52 | 6.82 | 8.75 | 26.08 |
| DeepSeek-R1-Distill-Qwen-14B | 57.16 | 62.05 | 21.36 | 38.52 | 17.16 | 34.89 | 8.30 | 21.70 | 32.64 |
| *Models Larger Than 20B Parameters* | | | | | | | | | |
| Gemma-3-27B | 82.27 | 78.07 | 50.91 | 47.16 | 39.09 | 35.00 | 19.43 | 32.39 | 48.04 |
| Qwen-2.5-32B | 89.55 | 73.18 | 20.80 | 34.66 | 14.66 | 26.48 | 3.75 | 23.86 | 35.87 |
| DeepSeek-R1-Distill-Qwen-32B | 82.73 | 71.82 | 43.75 | 49.77 | 36.02 | 38.64 | 15.80 | 25.57 | 45.51 |
| OREAL-32B | 55.23 | 63.86 | 26.02 | 48.07 | 25.57 | 41.25 | 20.00 | 35.23 | 39.40 |
| QwQ-32B | 94.20 | 91.59 | 68.64 | 77.27 | 55.00 | 52.05 | 26.82 | 33.18 | 62.34 |
| LLaMA-3.1-70B | 37.61 | 44.43 | 13.07 | 18.41 | 5.80 | 10.23 | 6.36 | 8.98 | 18.11 |
| Qwen-2.5-72B | 85.45 | 79.55 | 34.43 | 47.84 | 17.27 | 46.93 | 10.00 | 29.89 | 43.92 |
| o4-mini | 83.64 | 73.75 | 63.52 | 63.75 | 43.98 | 58.41 | 29.09 | 49.32 | 58.18 |
| DeepSeek-R1 | **95.80** | **94.55** | **78.75** | **86.59** | **57.95** | **74.20** | **47.61** | **57.61** | **74.13** |

# D Realistic vs Synthetic Multi-Needle Reasoning Tasks

In this section, we present a comparative analysis of model performance on realistic versus synthetic Multi-Needle Reasoning tasks. Our initial approach utilized realistic tasks based on real-world data from Wikipedia-based datasets (Inoue et al., 2020). However, such realistic benchmarks face the challenge of potential data contamination: once task sets are released, model developers may inadvertently or intentionally include these data in pretraining. This makes it difficult to fairly assess true reasoning ability, as high performance may simply reflect memorization rather than genuine reasoning capability. To address this limitation, we developed a synthetic task design for Multi-Needle-Reasoning tasks. These synthetic tasks are generated to match the structure, scale, and reasoning complexity of the original realistic tasks, but crucially, each instance is newly synthesized and does not have a fixed answer that could be memorized. This ensures that models cannot rely on memorization and must genuinely perform the intended reasoning.

Table 9 presents the performance of Qwen2.5 models on both realistic (v1) and synthetic (v2) multi-needle reasoning tasks. The results demonstrate a dramatic performance gap, with models achieving very high scores on realistic tasks but experiencing substantial drops (often 50-80

The substantial performance gaps observed in Tab. 9 indicate that realistic benchmarks may already be saturated and mainly reflect memorization rather than genuine reasoning capability. The consistently high performance on realistic tasks (often >90While larger models show better absolute performance on synthetic tasks, the relative performance drops remain substantial across all model sizes. These findings demonstrate that synthetic tasks, though less "realistic" in content, are necessary for valid and robust evaluation of reasoning ability, as they avoid the confounding effect of memorization and better reflect the true capabilities of large language models in multi-hop reasoning over long contexts.

Table 9: Performance Comparison of Qwen2.5 Models on Realistic vs Synthetic Multi-Needle Reasoning Tasks. v1: realistic tasks (Wikipedia-based), v2: synthetic tasks. $\Delta = v2 - v1$.

| Task | Qwen2.5-7B | | | Qwen2.5-14B | | | Qwen2.5-32B | | | Qwen2.5-72B | | |
|---|---|---|---|---|---|---|---|---|---|---|---|---|
| | v1 | v2 | $\Delta$ | v1 | v2 | $\Delta$ | v1 | v2 | $\Delta$ | v1 | v2 | $\Delta$ |
| Multi-Needle-Reasoning | 88.5 | 16.9 | -71.6 | 93.2 | 26.1 | -67.1 | 93.3 | 35.9 | -57.5 | 94.3 | 43.9 | -50.4 |
| Multi-Needle-Reasoning (EN) | 86.7 | 23.1 | -63.6 | 93.0 | 22.9 | -70.0 | 94.4 | 39.5 | -54.9 | 94.2 | 51.0 | -43.1 |
| Multi-Needle-Reasoning (ZH) | 90.3 | 10.7 | -79.7 | 93.4 | 29.2 | -64.2 | 92.2 | 32.2 | -60.0 | 94.5 | 36.8 | -57.7 |
| 2-Needle (EN) | 89.3 | 44.4 | -44.8 | 96.7 | 44.3 | -52.3 | 96.4 | 73.2 | -23.2 | 98.9 | 79.5 | -19.4 |
| 2-Needle (ZH) | 93.3 | 25.7 | -67.6 | 99.9 | 75.9 | -24.0 | 99.3 | 89.5 | -9.8 | 98.8 | 85.5 | -13.3 |
| 3-Needle (EN) | 86.3 | 26.6 | -59.7 | 90.1 | 20.2 | -69.8 | 88.7 | 34.7 | -54.0 | 89.2 | 47.8 | -41.4 |
| 3-Needle (ZH) | 95.6 | 7.6 | -88.0 | 90.2 | 19.0 | -71.2 | 89.5 | 20.8 | -68.8 | 98.0 | 34.4 | -63.5 |
| 4-Needle (EN) | 82.9 | 11.0 | -71.9 | 87.9 | 18.5 | -69.4 | 93.4 | 26.5 | -66.9 | 89.0 | 46.9 | -42.1 |
| 4-Needle (ZH) | 94.0 | 6.0 | -88.0 | 98.3 | 15.1 | -83.2 | 97.3 | 14.7 | -82.6 | 98.0 | 17.3 | -80.7 |
| 5-Needle (EN) | 88.2 | 10.4 | -77.8 | 97.3 | 8.8 | -88.5 | 99.3 | 23.9 | -75.4 | 99.5 | 29.9 | -69.7 |
| 5-Needle (ZH) | 78.5 | 3.4 | -75.1 | 85.2 | 6.8 | -78.4 | 82.7 | 3.8 | -79.0 | 83.4 | 10.0 | -73.4 |

# E   ATC Data Generation Algorithm

The ATC task employs a systematic algorithmic approach to generate synthetic family relationship datasets. The generation process ensures both linguistic diversity and logical consistency across different question types and complexity levels. We provide the detailed algorithm in Algorithm 1.

The algorithm generates each ATC instance through six key steps: (1) randomly sampling $n + 1$ unique names to form a sequential family chain, (2) assigning relationship terms with corresponding generation weights (1 for parent-child, 2 for grandparent-grandchild), (3) generating diverse relationship descriptions using predefined linguistic templates, (4) shuffling all relationship statements to eliminate positional bias, (5) creating question prompts based on the specified question type, and (6) computing ground truth answers according to the constructed family structure.

The generation process ensures comprehensive evaluation through multiple diversity mechanisms: name randomization prevents memorization, relationship template variation creates linguistic diversity, four question types evaluate different reasoning aspects, needle counts from 2 to 512 provide scalable complexity, and bilingual support enables cross-language evaluation. This algorithmic approach produces unique synthetic reasoning challenges that require genuine multi-step logical reasoning over information-dense contexts.

# F   Output Format Compliance Analysis

In this section, we address the potential bias caused by instruction-following errors, specifically the use of the \boxed{...} format required in our evaluation. We have carefully annotated and manually checked the outputs of the two smaller models, Qwen1.5-1.8B-Chat and Qwen2.5-1.5B-Instruct, as these are the only models where such errors were observed.

Table 10 presents the detailed results for these two models. The "false error rate" is defined as the proportion of cases where the model's answer is actually correct (as verified by human annotation), but is marked as wrong only because the output does not follow the required \boxed{...} format (i.e., an instruction-following error). We find that such errors are very rare: they only occur in these two small models, and only in the simplest setting with a single needle (needle count = 1). When the number of needles increases, the errors are no longer due to instruction-following, but rather because the models cannot solve the more complex reasoning task itself. Therefore, this represents a minor issue that only affects a very small subset of cases (small models, single-needle setting).

---

**Algorithm 1** ATC Data Generation Algorithm

---

**Require:** Number of needles $n$, Language $L \in \{\text{English}, \text{Chinese}\}$, Question type $Q$
**Ensure:** Generated prompt $P$ and ground truth answer $A$
1: $\mathcal{N} \leftarrow$ Random sample of $n+1$ unique names from name pool
2: $\mathcal{N} = \{name_1, name_2, \ldots, name_{n+1}\}$          $\triangleright$ Sequential chain structure
3: $\mathcal{R} \leftarrow \emptyset$          $\triangleright$ Relationship statements
4: $total\_gen \leftarrow 0$          $\triangleright$ Total generation weight
5: **for** $i = 1$ to $n$ **do**
6:      $rel\_term \leftarrow$ Random relationship term from $L$ vocabulary
7:      $gen\_weight \leftarrow$ Generation weight of $rel\_term$          $\triangleright$ 1 or 2
8:      $template \leftarrow$ Random template from $L$ patterns
9:      $statement \leftarrow template(name_i, name_{i+1}, rel\_term)$
10:      $\mathcal{R} \leftarrow \mathcal{R} \cup \{statement\}$
11:      $total\_gen \leftarrow total\_gen + gen\_weight$
12: **end for**
13: $\mathcal{R}_{shuffled} \leftarrow$ Randomly shuffle $\mathcal{R}$          $\triangleright$ Eliminate positional bias
14: $context \leftarrow$ Join $\mathcal{R}_{shuffled}$ into continuous text
15: **if** $Q = \text{ELDEST\_ANCESTOR}$ **then**
16:      $A \leftarrow name_1$
17:      $P \leftarrow$ Generate prompt asking for eldest ancestor of $name_{n+1}$
18: **else if** $Q = \text{NTH\_ANCESTOR}$ **then**
19:      $A \leftarrow name_1$
20:      $P \leftarrow$ Generate prompt asking for $total\_gen$-th ancestor of $name_{n+1}$
21: **else if** $Q = \text{NTH\_DESCENDANT}$ **then**
22:      $A \leftarrow name_{n+1}$
23:      $P \leftarrow$ Generate prompt asking for $total\_gen$-th descendant of $name_1$
24: **else if** $Q = \text{RELATIONSHIP\_DISTANCE}$ **then**
25:      $A \leftarrow total\_gen$
26:      $P \leftarrow$ Generate prompt asking for distance between $name_1$ and $name_{n+1}$
27: **end if**
28: $P \leftarrow$ Combine $context$ with question prompt
29: **return** $P, A$

---

Table 10: False error rate (%) caused by instruction-following errors (\boxed{...} format) in small models.

| Needles | Qwen1.5-1.8B-Chat | Qwen2.5-1.5B-Instruct |
|---|---|---|
| 1 | 25.0 | 12.5 |
| 2 | 0.0 | 0.0 |
| 4 | 0.0 | 0.0 |
| 8 | 0.0 | 0.0 |
| 16 | 0.0 | 0.0 |
| 32 | 0.0 | 0.0 |
| 64 | 0.0 | 0.0 |
| 128 | 0.0 | 0.0 |
| 256 | 0.0 | 0.0 |
| 512 | 0.0 | 0.0 |

## G *NeedleBench* Prompt Examples

This section presents representative prompt examples for each major task in *NeedleBench*.

---

**Single-Needle Retrieval (Needle First - Demonstration with English Version)**

**Prompt:**

This is a test of long-text capability. You need to first read the long document below, and then answer the final question based on the information in the document.
The content of the long document is as follows

**Hidden on Emerald Island is the legendary Stardust Shard.**
*—Paul Graham Essays— —Paul Graham Essays— —Paul Graham Essays— —Paul Graham Essays— —Paul Graham Essays— —Paul Graham Essays— —Paul Graham Essays— —Paul Graham Essays— —Paul Graham Essays—*
*—Paul Graham Essays— —Paul Graham Essays— —Paul Graham Essays— —Paul Graham Essays— —Paul Graham Essays— —Paul Graham Essays— —Paul Graham Essays— —Paul Graham Essays— —Paul Graham Essays—*

Based on the information in the document, now please answer: What legendary item is hidden on Emerald Island? Please answer in the format 'The legendary item hidden on the Emerald Island is _________.'

---

Figure 7: An example prompt of Single-Needle Retrieval showcasing key information with the single needle positioned at the very beginning. In actual tests, the needle is placed at various depths within extended texts to evaluate performance under different conditions.

---

**Single-Needle Retrieval (Needle Middle - Demonstration with English Version)**

**Prompt:**
This is a test of long-text capability. You need to first read the long document below, and then answer the final question based on the information in the document.
The content of the long document is as follows

*—Paul Graham Essays— —Paul Graham Essays— —Paul Graham Essays— —Paul Graham Essays— —Paul Graham Essays— —Paul Graham Essays— —Paul Graham Essays— —Paul Graham Essays— —Paul Graham Essays—*
**Hidden on Emerald Island is the legendary Stardust Shard.**
*—Paul Graham Essays— —Paul Graham Essays— —Paul Graham Essays— —Paul Graham Essays— —Paul Graham Essays— —Paul Graham Essays— —Paul Graham Essays— —Paul Graham Essays— —Paul Graham Essays—*

Based on the information in the document, now please answer: What legendary item is hidden on Emerald Island? Please answer in the format 'The legendary item hidden on the Emerald Island is _________.'

---

Figure 8: An example prompt of Single-Needle Retrieval showcasing key information with the single needle positioned at the middle

## G.1 Single-Needle Retrieval

Figure 7, Fig. 8, and Fig. 9 show prompt examples for the Single-Needle Retrieval task, where the target information (needle) is placed at different positions within the context (beginning, middle, and end, respectively).

---

**Single-Needle Retrieval (Needle Last - Demonstration with English Version)**

**Prompt:**
This is a test of long-text capability. You need to first read the long document below, and then answer the final question based on the information in the document.
The content of the long document is as follows

*—Paul Graham Essays— —Paul Graham Essays— —Paul Graham Essays— —Paul Graham Essays— —Paul Graham Essays— —Paul Graham Essays— —Paul Graham Essays— —Paul Graham Essays— —Paul Graham Essays—*
*—Paul Graham Essays— —Paul Graham Essays— —Paul Graham Essays— —Paul Graham Essays— —Paul Graham Essays— —Paul Graham Essays— —Paul Graham Essays— —Paul Graham Essays— —Paul Graham Essays—*
**Hidden on Emerald Island is the legendary Stardust Shard.**

Based on the information in the document, now please answer: What legendary item is hidden on Emerald Island? Please answer in the format 'The legendary item hidden on the Emerald Island is __________.'

---

Figure 9: An example prompt of Single-Needle Retrieval showcasing key information with the single needle positioned at last

### G.2 Multi-Needle Retrieval

Figure 10 provides a prompt example for the Multi-Needle Retrieval task, which requires the model to extract multiple target items from a long context.

### G.3 Multi-Needle Reasoning

Figure 11 presents a prompt example for the Multi-Needle Reasoning task, where the model must perform reasoning over several interrelated pieces of information distributed throughout the context.

### G.4 Ancestral Trace Challenge

Figure 12 shows a prompt example for the Ancestral Trace Challenge (ATC), an information-dense task that requires multi-step logical reasoning to trace relationships and infer the correct answer.

## H  Error Analysis Examples

In this section, we present representative error cases observed in the Ancestral Trace Challenge (ATC) task. Each error type is illustrated with a concrete example and referenced in the main text.

### H.1  Under-thinking Error

See Fig. 13 for an example of an Under-thinking Error. This error occurs when the model prematurely halts its reasoning process, incorrectly concluding that no further inference can be made, even though additional relevant clues remain in the context. As a result, the model fails to identify the true eldest ancestor.

### H.2  Instruction Following Error

See Fig. 14 for an example of an Instruction Following Error. In this case, the model's reasoning may be correct, but it fails to adhere to the required output format specified in the task instructions (e.g., omitting the \boxed{} format), resulting in an incomplete or invalid response.

---

**Multi-Needle Retrieval (Demonstration with five Needles English Version Prompt)**

**Prompt:**
This is a test of long-text capability. You need to first read the long document below, and then answer the final questions one by one based on the information in the document.
The content of the long document is as follows

*—Paul Graham Essays— —Paul Graham Essays— —Paul Graham Essays—*
**Hidden on Forgotten Island is the legendary Stardust Shard.**
*—Paul Graham Essays— —Paul Graham Essays— —Paul Graham Essays—*
**Hidden on Mythical Island is the legendary Time-Space Key.**
*—Paul Graham Essays— —Paul Graham Essays— —Paul Graham Essays—*
**The ruler of the Alpha Bot star system is Cosmic Ruler Starshine.**
*—Paul Graham Essays— —Paul Graham Essays— —Paul Graham Essays—*
**Hidden on Storm Island is the legendary Goodness Heart.**
*—Paul Graham Essays— —Paul Graham Essays— —Paul Graham Essays—*
**The ruler of the Orion star system is Guardian of Time Lightspeed.**
*—Paul Graham Essays— —Paul Graham Essays— —Paul Graham Essays—*

Based on the information in the document, now please answer: What legendary item is hidden on Forgotten Island? What legendary item is hidden on Mythical Island? Who is the ruler of the Alpha Bot star system? What legendary item is hidden on Storm Island? Who is the ruler of the Orion star system? Please answer in the format of "The legendary item hidden on the Forgotten Island is _________, The legendary item hidden on the Mythical Island is _________, The ruler of the Alpha Bot star system is _________, The legendary item hidden on the Storm Island is _________, The ruler of the Orion star system is _________."

Figure 10: An example prompt of Multi-Needle Retrieval with the new question template (5 needles)

### H.3 Partial Understanding Error

See Fig. 15 for an example of a Partial Understanding Error. This error type is characterized by the model identifying only a subset of the relationships or information present in the context, leading to an incomplete or partially correct answer.

### H.4 Repetitive Output Error

See Fig. 16 for an example of a Repetitive Output Error. Here, the model either enters a repetitive reasoning loop—reiterating the same inference steps or chains without reaching a conclusion—or repeatedly outputs meaningless or irrelevant content.

### H.5 Hallucination Error

See Fig. 17 for an example of a Hallucination Error. This error occurs when the model introduces information or relationships that are not present in the original context, resulting in fabricated or unsupported answers.

---

**Multi-Needle Reasoning (Demonstration with Three Needles English Version)**

**Prompt:**

This is a test of long-text capability. You need to first read the long document below, and then answer the final question based on the information in the document.

The content of the long document is as follows

*—Paul Graham Essays— —Paul Graham Essays— —Paul Graham Essays—*

Jasmine Lane is not only James Hill's father but also James Hill's role model.

*—Paul Graham Essays— —Paul Graham Essays— —Paul Graham Essays—*

Janet Guzman is not only Carolyn Hicks's maternal grandmother but also Carolyn Hicks's role model.

*—Paul Graham Essays— —Paul Graham Essays— —Paul Graham Essays—*

James Hill, as Janet Guzman's paternal grandfather, has a significant impact on Janet Guzman's upbringing.

*—Paul Graham Essays— —Paul Graham Essays— —Paul Graham Essays—*

Based on the information in the document, now please answer: Given the context described above, who is the eldest relative that 'Carolyn Hicks' can trace back to in the context?

For example:

Example 1: If James Hill's father is Jasmine Lane, and no further information about familial relationships is provided in the text, then the oldest relative James Hill can trace back to in the provided text is \boxed{Jasmine Lane}.

Example 2: If Andrew Williams's grandmother is Dan Newton, and Dan Newton's father is James Hill, and no further information about familial relationships is provided in the text, then the oldest relative Andrew Williams can trace back to in the provided text is \boxed{James Hill}.

Example 3: If Jeff White's father is Kevin Le, Dan Newton's grandmother is Jeff White, and Jeff White's father is Kevin Le, and Shelley Mills is Dan Newton's great-granddaughter, and no further information about familial relationships is provided in the text, then the oldest relative Shelley Mills can trace back to in the provided text is \boxed{Kevin Le}.

Notes:

1. You do not need to worry about the gender consistency of names in this test. For example, a name that is typically considered feminine can still be the father of another person. Our primary focus is on who is older.

2. Ignore surname inheritance issues. For instance, Andrew Williams could still be the biological father of Christopher Baker. We only care about who is older and do not need to consider whether a child should inherit the father's or mother's surname.

3. At the end of your response, remember to put your final answer within \boxed{}. For example: "So the oldest relative 'Carolyn Hicks' can trace back to in the provided text is \boxed{(your answer here)}."

---

Figure 11: An example prompt of Multi-Needle Reasoning

---

**Ancestral Trace Challenge (3-needle version example with DeepSeek R1 response)**

**Prompt:**
Here is a test for multi-step reasoning ability called the Ancestral Trace Challenge. In this test, we will simulate different people's familial relationships, and your task is to continuously reason through them until you identify the eldest ancestor.
Now, the scrambled family relationships are provided below:
Wyatt James is the child of Maria Watson. Emily Barry, as Maria Watson's paternal grandfather, has a significant impact on Maria Watson's upbringing. Joseph Taylor is not only Emily Barry's maternal grandfather but also Emily Barry's role model.

Given the scrambled family relationships described above, who is the eldest relative that 'Wyatt James' can trace back to in the context?

For example:
Example 1: If James Hill's father is Jasmine Lane, and no further information about familial relationships is provided in the text, then the oldest relative James Hill can trace back to in the provided text is \boxed{Jasmine Lane}.
Example 2: If Andrew Williams's grandmother is Dan Newton, and Dan Newton's father is James Hill, and no further information about familial relationships is provided in the text, then the oldest relative Andrew Williams can trace back to in the provided text is \boxed{James Hill}.
Example 3: If Jeff White's father is Kevin Le, Dan Newton's grandmother is Jeff White, and Jeff White's father is Kevin Le, and Shelley Mills is Dan Newton's great-granddaughter, and no further information about familial relationships is provided in the text, then the oldest relative Shelley Mills can trace back to in the provided text is \boxed{Kevin Le}.

Notes:
1. You do not need to worry about the gender consistency of names in this test. For example, a name that is typically considered feminine can still be the father of another person. Our primary focus is on who is older.
2. Ignore surname inheritance issues. For instance, Andrew Williams could still be the biological father of Christopher Baker. We only care about who is older and do not need to consider whether a child should inherit the father's or mother's surname.
3. At the end of your response, remember to put your final answer within \boxed{}. For example: "So the oldest relative xxx(the person you need to find the eldest relative for) can trace back to in the provided text is \boxed{}."

**DeepSeek R1 Response (Correct Example):**
Wyatt James is the child of Maria Watson, making Maria one generation above Wyatt. Maria's paternal grandfather is Emily Barry, placing Emily two generations above Maria and three generations above Wyatt. Emily Barry's maternal grandfather is Joseph Taylor, which adds two more generations above Emily (Emily's mother and then Joseph). Therefore, Joseph Taylor is five generations above Wyatt. Among the mentioned relatives, Joseph Taylor is the eldest ancestor.

So the oldest relative Wyatt James can trace back to in the provided text is \boxed{Joseph Taylor}.

---

Figure 12: Example prompt and DeepSeek R1 response in the Ancestral Trace Challenge (ATC) with multi-step reasoning

---

**Ancestral Trace Challenge (Under-thinking Error Example by GPT-4.1)**

**Prompt:**
***The initial question prompt is omitted here for brevity; see Fig. 14 for identical content.***
Now, the scrambled family relationships are provided below:
James Tate's father is Natasha Weeks. Gary Anderson is not only Casey Johnson's father but also Casey Johnson's role model. For Cesar Pacheco, Dana Martinez is not just a mom, but also a friend. Taylor Collins is the child of Douglas Harris. Casey Johnson is not only Natasha Weeks's maternal grandfather but also Natasha Weeks's role model. Daniel Morton is not only Jessica Stewart's mom but also Jessica Stewart's role model. For John Cole MD, Sandra Cameron is more than just a father; Sandra Cameron is a lifelong mentor of John Cole MD. Tami Woodward, as Sandra Cameron's grandmother, has a significant impact on Sandra Cameron's upbringing. Mary Smith is William Castaneda's dad. Stephen Hunter's maternal grandmother is James Tate. For Kevin Le, Nicole Downs is not just a paternal grandmother, but also a friend. Dennis Middleton is Michael Martinez's mother. Kelly Meyer's mom is Eileen Green. For Sheri Garrett, Holly Jones is more than just a paternal grandmother; Holly Jones is a lifelong mentor of Sheri Garrett. For Christy Dixon, Cesar Pacheco is not just a maternal grandmother, but also a friend. Gary Anderson's father is John Cole MD. Sheri Garrett is Kaitlyn Chandler's maternal grandmother. Jonathan Payne is the child of Julie Ross. Jessica Martin is the child of Shannon Garner. Julie Ross is the child of Taylor Collins. For Samantha Sheppard, Kaitlyn Chandler is more than just a mother; Kaitlyn Chandler is a lifelong mentor of Samantha Sheppard. For Nicole Downs, Jessica Stewart is not just a mother, but also a friend. Kelly Meyer, as Mary Smith's father, has a significant impact on Mary Smith's upbringing. For Shannon Garner, Stephen Hunter is not just a mother, but also a friend. Christy Dixon is Sean Mccoy's grandfather. Jonathan Payne is not only Daniel Morton's mother but also Daniel Morton's role model. Holly Jones's paternal grandfather is Michael Martinez. Douglas Harris is the child of William Castaneda. **Jessica Martin is Eileen Green's paternal grandmother.** Samantha Sheppard, as Dana Martinez's mom, has a significant impact on Dana Martinez's upbringing. Tami Woodward's maternal grandmother is Sean Mccoy.

Given the scrambled family relationships described above, who is the eldest relative that 'Kevin Le' can trace back to in the context?

**GPT-4.1 Response (Under-thinking Error):**
Let's break down the relationships relevant to **Kevin Le** by tracing up along every provided parental and grandparental link. We are looking for the furthest ancestor **Kevin Le** can trace from the data.
***Step 1−11 are all correctly reasoned and omitted here due to space constraints.***

Step 12: Does Eileen Green have an ancestor?
Look for Eileen Green as a child in any other statement. Let's check all the family chain we have so far.

No further information about Eileen Green's parentage appears in the provided text.

*The model's response summary here is omitted to highlight the subsequent error in its answer.*

**Final answer:**
So, the eldest relative 'Kevin Le' can trace back to in the provided text is \boxed{Eileen Green}.

---

Figure 13: An example of an Under-thinking Error in the Ancestral Trace Challenge (ATC) task. The GPT-4.1 model prematurely concludes the reasoning process and fails to identify the true eldest ancestor, despite the presence of additional relevant information in the context.

---

**Ancestral Trace Challenge (Instruction Following Error Example by Qwen1.5-1.8B-Chat)**

**Prompt:**
Here is a test for multi-step reasoning ability called the Ancestral Trace Challenge. In this test, we will simulate different people's familial relationships, and your task is to continuously reason through them until you identify the eldest ancestor.
For example:
Example 1: If James Hill's father is Jasmine Lane, and no further information about familial relationships is provided in the text, then the oldest relative James Hill can trace back to in the provided text is \boxed{Jasmine Lane}.
Example 2: If Andrew Williams's grandmother is Dan Newton, and Dan Newton's father is James Hill, and no further information about familial relationships is provided in the text, then the oldest relative Andrew Williams can trace back to in the provided text is \boxed{James Hill}.
Example 3: If Jeff White's father is Kevin Le, Dan Newton's grandmother is Jeff White, and Jeff White's father is Kevin Le, and Shelley Mills is Dan Newton's great-granddaughter, and no further information about familial relationships is provided in the text, then the oldest relative Shelley Mills can trace back to in the provided text is \boxed{Kevin Le}.

Notes:
1. You do not need to worry about the gender consistency of names in this test. For example, a name that is typically considered feminine can still be the father of another person. Our primary focus is on who is older.
2. Ignore surname inheritance issues. For instance, Andrew Williams could still be the biological father of Christopher Baker. We only care about who is older and do not need to consider whether a child should inherit the father's or mother's surname.
3. At the end of your response, remember to put your final answer within \boxed{}. For example: "So the oldest relative 'Andrew Williams' can trace back to in the provided text is \boxed{somebody (your answer here)}."

Now, the scrambled family relationships are provided below:
For Andrew Williams, Dan Newton is more than just a mother; Dan Newton is a lifelong mentor of Andrew Williams.

Given the scrambled family relationships described above, who is the eldest relative that 'Andrew Williams' can trace back to in the context?

**Qwen1.5-1.8B-Chat Response (Instruction Following Error):**
Based on the scrambled family relationships provided, the eldest relative that 'Andrew Williams' can trace back to is Dan Newton. This is because Dan Newton is mentioned as Andrew's grandmother, and no further information about familial relationships is provided in the text. Therefore, we can infer that Dan Newton is the oldest relative and the one who has been a mentor to Andrew throughout his life.

---

Figure 14: An example of an Instruction Following Error in the ATC task. The Qwen1.5-1.8B-Chat model gives the correct reasoning but fails to output the answer in the required \boxed{} format.

---

**Ancestral Trace Challenge (Partial Understanding Error Example by InternLM3-8B)**

**Prompt:**

***The initial question prompt is omitted here for brevity; see Fig. 14 for identical content.***

Now, the scrambled family relationships are provided below:

**For Andrew Williams, Dan Newton is more than just a mother; Dan Newton is a lifelong mentor of Andrew Williams.** Andrew Williams, as Brooke Sanders's mother, has a significant impact on Brooke Sanders's upbringing. Brooke Sanders is not only Leah Williams's maternal grandfather but also Leah Williams's role model.

Given the scrambled family relationships described above, who is the eldest relative that 'Leah Williams' can trace back to in the context?

**InternLM3-8B Response (Partial Understanding Error):**

To determine the eldest relative that Leah Williams can trace back to, let's break down the given family relationships step by step:

1. **Andrew Williams and Dan Newton:**
- **Dan Newton is a lifelong mentor of Andrew Williams. This does not imply a familial relationship, so we can ignore this for our purpose of tracing ancestry.**

2. **Andrew Williams and Brooke Sanders:**
- Andrew Williams is Brooke Sanders's mother. This means Andrew Williams is the mother of Brooke Sanders.

3. **Brooke Sanders and Leah Williams:**
- Brooke Sanders is Leah Williams's maternal grandfather. This means Brooke Sanders is the father of Leah Williams's mother.

From these relationships, we can trace the lineage as follows:
- Leah Williams's mother is Andrew Williams.
- Andrew Williams's father is Brooke Sanders.

Therefore, the eldest relative that Leah Williams can trace back to in the provided text is \boxed{Brooke Sanders}.

---

Figure 15: An example of a Partial Understanding Error in the ATC task. The model locates the key information but only interprets part of it—for example, it notices "Dan Newton is more than just a mother; Dan Newton is a lifelong mentor of Andrew Williams" but only treats Dan Newton as a mentor, missing the familial relationship.

---

**Ancestral Trace Challenge (Repetitive Output Error Example by Deepseek-R1-Distill-Qwen-7B)**

**Prompt:**
*The initial question prompt is omitted here for brevity; see Fig. 14 for identical content.*
Now, the scrambled family relationships are provided below:
Keith Jordan is the child of Anna Ford. Anna Ford's grandfather is Brian Barron. Kayla Lucas is Brian Barron's mom.

Given the scrambled family relationships described above, who is the eldest relative that 'Keith Jordan' can trace back to in the context?

**Deepseek-R1-Distill-Qwen-7B Response (Repetitive Output Error):**
Okay, so I need to figure out who is the eldest ancestor that Keith Jordan can trace back to based on the given family relationships. Let me break down the information step by step.

First, the information provided is:
1. Keith Jordan is the child of Anna Ford.
2. Anna Ford's grandfather is Brian Barron.
3. Kayla Lucas is Brian Barron's mom.

I need to determine the eldest relative Keith can trace back to. Let's start by mapping out the family tree based on the given information.

Starting with Keith Jordan, he has a parent, Anna Ford. So, Anna is Keith's mother. Now, Anna's grandfather is Brian Barron. That means Brian is two generations above Anna. Since Anna is Keith's mother, Brian would be the great-grandfather of Keith. Because Brian is the grandfather of Anna, who is Keith's mother, Brian is the great-grandfather of Keith.

Additionally, Kayla Lucas is Brian Barron's mom. So, Kayla is Brian's daughter. Since Brian is the great-grandfather of Keith, Kayla would be the great-grandmother of Keith. **But wait,** Kayla is Brian's mom, so she is three generations above Keith. That makes her the great-grandmother, not the grandmother.

**Wait, let me clarify.** If Brian is the grandfather of Anna, then Anna is Brian's granddaughter. Since Anna is Keith's mother, Keith is Brian's great-grandson. Therefore, Brian is the great-grandfather of Keith. Kayla is Brian's mother, so she is three generations above Keith, making her the great-grandmother.

But the question is asking for the eldest relative that Keith can trace back to. So, starting from Keith, the immediate parent is Anna, then Anna's parent is Brian, and Brian's parent is Kayla. So, the chain is Keith → Anna → Brian → Kayla.

Therefore, the eldest relative Keith can trace back to is Kayla Lucas, who is his great-grandmother.

*[The model continues to repeat the same reasoning and chain multiple times, restating the relationships and the conclusion in nearly identical wording, e.g., "Wait, but in the examples, when someone's grandmother is mentioned, the eldest is the father of that grandmother..." and so on, for several more paragraphs.]*

---

Figure 16: An example of a Repetitive Output Error in the ATC task. The Deepseek-R1-Distill-Qwen-7B model repeatedly tries to apply the example pattern of tracing to a grandparent's parent, but since no further ancestor is given for Kayla Lucas, it gets stuck in a loop, restating the same reasoning chain over and over.

---

**Ancestral Trace Challenge (Hallucination Error Example by Deepseek-R1-Distill-Qwen-7B)**

**Prompt:**
***The initial question prompt is omitted here for brevity; see Fig. 14 for identical content.***
Now, the scrambled family relationships are provided below:
For Nancy Gilbert, Kathy Marshall is not just a maternal grandmother, but also a friend. For Carol Barron, George Estes is not just a mom, but also a friend. Carol Barron is not only Kathy Marshall's maternal grandfather but also Kathy Marshall's role model.

Given the scrambled family relationships described above, who is the eldest relative that 'Nancy Gilbert' can trace back to in the context?

**Deepseek-R1-Distill-Qwen-7B Response (Hallucination Error):**
Alright, let's try to figure out who the eldest ancestor Nancy Gilbert can trace back to. I'm a bit confused, but I'll take it step by step.
First, I need to understand the given information. The problem provides some scrambled family relationships:
1. For Nancy Gilbert, Kathy Marshall is not just a maternal grandmother but also a friend.
2. For Carol Barron, George Estes is not just a mom but also a friend.
3. Carol Barron is not only Kathy Marshall's maternal grandfather but also her role model.

*The model's reasoning process here is omitted to highlight the subsequent error in its answer.*

**Additionally, Carol is the mother of George Estes, who is a friend but not necessarily related to Nancy.** So, the key here is that Carol is the maternal grandfather of Kathy, who is the maternal grandmother of Nancy. Therefore, Carol is the great-grandfather of Nancy's mother, making him the great-great-grandfather of Nancy.
But wait, the question is about the eldest ancestor Nancy can trace back to. So, starting from Nancy, her maternal grandmother is Kathy, whose parent is her maternal great-grandmother, and her grandfather is Carol. So, Carol is the great-grandfather of Nancy's mother, which makes him the great-great-grandfather of Nancy. Therefore, Carol is the eldest ancestor Nancy can trace back to.
Wait, but let me double-check.
Nancy's maternal grandmother is Kathy. Kathy's parent is Nancy's maternal great-grandmother. Kathy's father is Carol, making Carol the maternal grandfather of Nancy's mother. Therefore, Carol is the great-grandfather of Nancy's mother, which makes him the great-great-grandfather of Nancy. So, Carol is the eldest ancestor Nancy can trace back to.
I think that's it. So, the answer should be Carol Barron.
So the eldest relative Nancy Gilbert can trace back to in the provided text is \boxed{Carol Barron}.

---

Figure 17: An example of a Hallucination Error in the ATC task. The Deepseek-R1-Distill-Qwen-7B model hallucinates the relationship in "For Carol Barron, George Estes is not just a mom, but also a friend," mistakenly treating Carol as the mother of George Estes and thus inferring the wrong ancestor.

