# OpenReview forum: "NeedleBench: Evaluating LLM Retrieval and Reasoning Across Varying Information Densities"
_TMLR — Accepted by TMLR_

### Review · Reviewer_oHUd · 2025-06-18

**Summary Of Contributions:**

This paper presents a comprehensive benchmark, NeedleBench, for evaluating LLMs on long-context retrieval and reasoning tasks. The authors cover two categories of tasks, e.g., information-sparse and information-dense, to simulate a wide range of real-world scenarios. The benchmark reveals critical limitations of state-of-the-art models, particularly in dense reasoning tasks, and identifies under-thinking as a key failure mode in long-context inference.

**Audience:**

Yes

**Claims And Evidence:**

Yes

**Requested Changes:**

* Consider revising the title to reflect the dual focus on both sparse and dense information contexts.
* Expand the benchmark to include more open and closed-source models in the information-sparse evaluation
* Include statistics on how often models fail to follow output format instructions (e.g., missing \boxed{...}), and discuss whether this may affects model performance

**Strengths And Weaknesses:**

### Strengths

* The paper is well written and clearly structured
* The proposed benchmark, NeedleBench, is technically sound and comprehensively covers both information-sparse and information-dense scenarios.
* The experimental analysis is thorough, with detailed breakdowns by task type, model size, and needle depth.

### Weaknesses
* The benchmark currently includes only a limited subset of open-source models for the information-sparse tasks. Including widely-used models such as DeepSeek distill models and some close source models would strengthen the comparative analysis.
* The Single-Needle Retrieval task may be too easy for current LLMs and fails to discriminate among stronger models, as many achieve near-perfect scores.
* The title does not fully reflect the benchmark's comprehensive scope, particularly the strong coverage of information-sparse tasks.
* Minor: The evaluation relies on the model returning an answer in a \boxed{...} format. It is unclear how often models fail due to instruction-following errors. This could bias results, especially for smaller models with poor instruction following ability.

---

> ### Author Response · Authors · 2025-07-14
>
> Thank you very much for your detailed and valuable feedback. Here are our responses:
>
> 1. **Benchmark Model Coverage:**
>
> We appreciate the suggestion to include additional widely-used models. In the original version, we mainly focused the Information-Sparse tasks on non-Long-CoT models due to both our experimental design and resource constraints. Initially, we focused on evaluating ​**Long-CoT models mainly on information-dense tasks**​, as these present greater challenges. However, following your suggestion, we have conducted supplementary experiments on several state-of-the-art Long-CoT models, including OpenAI's o4-mini, DeepSeek R1, and DeepSeek distill models, as well as some closed-source models. The results of these additional experiments have been included in Appendix B of the revised manuscript.
>
> **Table 1: Supplementary results on NeedleBench-128K**
>
> | Dataset                   | Single-Needle Retrieval-EN (128K) | Single-Needle Retrieval-ZH (128K) | Single-Needle Retrieval (128K) | Multi-Needle Retrieval-EN (128K) | Multi-Needle Retrieval-ZH (128K) | Multi-Needle Retrieval (128K) | Multi-Needle Reasoning-EN (128K) | Multi-Needle Reasoning-ZH (128K) | Multi-Needle Reasoning (128K) | NeedleBench Overall Score (128K) |
> | --------------------------- | ----------------------------------- | ----------------------------------- | -------------------------------- | ---------------------------------- | ---------------------------------- | ------------------------------- | ---------------------------------- | ---------------------------------- | ------------------------------- | ---------------------------------- |
> | DeepSeek-Distill-Qwen-7B  | 46.91                             | 41.95                             | 44.43                          | 46.91                            | 41.95                            | 44.43                         | 16.65                            | 10                               | 13.32                         | 34.06                            |
> | DeepSeek-Distill-Qwen-14B | 95.95                             | 94.68                             | 95.32                          | 95.95                            | 94.68                            | 95.32                         | 39.29                            | 25.99                            | 32.64                         | 74.43                            |
> | DeepSeek-Distill-Qwen-32B | 98.14                             | 97.18                             | 97.66                          | 98.14                            | 97.18                            | 97.66                         | 46.45                            | 44.57                            | 45.51                         | 80.28                            |
> | OREAL-32B                 | 96.82                             | 96.64                             | 96.73                          | 96.82                            | 96.64                            | 96.73                         | 47.1                             | 31.7                             | 39.4                          | 77.62                            |
> | QwQ-32B                   | 98.05                             | 98.5                              | 98.27                          | 98.05                            | 98.5                             | 98.27                         | 63.52                            | 61.16                            | 62.34                         | 86.3                             |
> | o4-mini                   | 99.14                             | 99.18                             | 99.16                          | 99.14                            | 99.18                            | 99.16                         | 61.31                            | 55.06                            | 58.18                         | 85.5                             |
> | Deepseek-R1               | 99.91                             | 99.32                             | 99.61                          | 99.91                            | 99.32                            | 99.61                         | 78.24                            | 70.03                            | 74.13                         | 91.12                            |
>
> 2. **Single-Needle Retrieval Task:**
>
> We acknowledge that the Single-Needle Retrieval task may seem relatively straightforward. However, it remains essential for evaluating a model’s fundamental retrieval abilities, particularly in ultralong context scenarios. For example, Table 1 shows that not all models, such as DeepSeek R1 distilled 7B do not consistently excel at this task, underscoring its ongoing relevance. Moreover, this task serves as a foundational step toward more complex tasks, such as multi-needle retrieval and reasoning. Therefore, we appreciate your feedback and will keep this task as a component of our benchmark.

---

> ### Author Response · Authors · 2025-07-14
>
> 3. **Title Revision:**
>
> We appreciate your suggestion to revise the title to better reflect the comprehensive scope of the benchmark. We propose updating the title as follows:
>
> > "NeedleBench: Evaluating LLM Retrieval and Reasoning Across Varying Information Densities"
>
> 4. **Output Format Compliance:**
>
> Thank you for your comments regarding the potential bias caused by instruction-following errors, specifically the use of the \\boxed{...} format. We have carefully annotated and manually checked the outputs of the two smaller models, Qwen1.5-1.8B-Chat and Qwen2.5-1.5B-Instruct, as these are the only models where such errors were observed. Table 2 below presents the detailed results for these two models. We also include relevant results in Appendix F of the revised manuscript.
>
> In Table 2, the "false error rate" is defined as the proportion of cases where the model's answer is actually correct (as verified by human annotation), but is marked as wrong only because the output does not follow the required \\boxed{...} format (i.e., an instruction-following error). We find that ​**such errors are very rare**​: they only occur in these two small models, and only in the simplest setting with a single needle (needle count = 1). When the number of needles increases, the errors are no longer due to instruction-following, but rather because the models cannot solve the more complex reasoning task itself. Therefore, we do not attribute those errors to instruction-following.
>
> As you have pointed out, we agree that this is a ​**minor issue**​, since it only affects a very small subset of cases (small models, single-needle setting). For completeness, we provide the detailed statistics in Table 2.
>
> **Table 2. False error rate (%) caused by instruction-following errors (\\boxed{...} format) in small models.**
>
> | Needles | Qwen1.5-1.8B-Chat | Qwen2.5-1.5B-Instruct |
> | --------- | ------------------- | ----------------------- |
> | 1       | 25.0              | 12.5                  |
> | 2       | 0.0               | 0.0                   |
> | 4       | 0.0               | 0.0                   |
> | 8       | 0.0               | 0.0                   |
> | 16      | 0.0               | 0.0                   |
> | 32      | 0.0               | 0.0                   |
> | 64      | 0.0               | 0.0                   |
> | 128     | 0.0               | 0.0                   |
> | 256     | 0.0               | 0.0                   |
> | 512     | 0.0               | 0.0                   |

---

### Review · Reviewer_q1AZ · 2025-06-27

**Summary Of Contributions:**

The paper introduces a benchmark designed to evaluate how well large language models (LLMs) retrieve and reason over long, information-rich contexts. It features bilingual (English-Chinese) tasks across adaptive context lengths (32k to 128k tokens), including both traditional information-sparse needle-in-a-haystack type tasks as well as more complex multi-needle reasoning and a new information-dense challenge called the Ancestral Trace Challenge (ATC), which requires reasoning over up to 512 needles distributed in $\leq$ 9.6k tokens.

The study finds that while many open source LLMs excel at sparse retrieval, they perform poorly in multi-step reasoning in both dense and sparse cases. Even expensive closed source LLMs like Claude 3.7-Sonnet-Thinking and OpenAI's o3-mini model struggle in the information dense ATC task, often halting prematurely (under-thinking). Evaluation is performed using a variant of the Exact Match (EM) metric and ENL, a measure of the largest number of needles for which the EM accuracy is greater than a threshold. The main conclusion is that while retrieval is largely solved, sustained reasoning remains a key limitation in current LLMs.

**Audience:**

Yes

**Claims And Evidence:**

Yes

**Requested Changes:**

1. Figure 1 is very cluttered and difficult to parse. Consider separating it into multiple figures.

2. What is meant by "instruction truncation" in the last sentence of section 3.1? Why would it be caused by tokenizer discrepancies? How does subtracting a buffer from the target context length help? Please explain more clearly

**Strengths And Weaknesses:**

Strengths:

1. Introduction of a comprehensive benchmark that evaluates LLMs on both sparse and dense information retrieval tasks and the introduction of a novel dense multi-needle reasoning task, the Ancestral Trace Challenge (ATC), where even the state of the art closed source reasoning LLMs are seen to struggle.
2. Effective Needle Length at 50% (ENL-50) is an insightful metric which captures the depth of reasoning in addition to just retrieval accuracy but also the depth of reasoning

Weaknesses:

1. Lack of realistic multi-needle reasoning tasks (ATC is a synthetic task which may not fairly reflect model performance in real world settings).
2. The data generation process is not clearly explained. It would be useful, especially for ATC, to know how the text is generated and the needles are inserted to get a sense of the scalability of the process and its generalizability to other tasks or domains.
3. It is not mentioned if the other benchmarks mentioned in Section 2 use the same evaluation metrics, nor are the evaluation insights compared with those obtained from other benchmarks. Thus, we have no way of verifying if this benchmark indeed is better at capturing the shortcomings of LLMs in reasoning and retrieval over long-context data.

---

> ### Author Response · Authors · 2025-07-14
>
> Thank you very much for your valuable feedback and suggestions. Below are detailed responses to your comments:
>
> 1. **Realistic Tasks v.s. Synthetic Tasks**
>
> We thank the reviewer for raising the important question about the realism of our multi-needle reasoning tasks. There is indeed a trade-off between using realistic tasks (which are closer to real-world applications) and synthetic tasks (which better control for the model’s prior knowledge and memorization from pretraining). In our previous version, we adopted realistic tasks based on real-world data, using the dataset from [1]. However, once the task set is released, it becomes possible for model developers to ​**saturate the benchmark by including these data in pretraining, either intentionally or unintentionally**​. This makes it difficult to fairly assess true reasoning ability, as high performance may simply reflect memorization rather than genuine reasoning.
>
> To address this, in our current version, we have adopted a more flexible synthetic task design. The synthetic tasks are generated to match the structure, scale, and reasoning complexity of the original realistic tasks, but crucially, each instance is newly synthesized and does not have a fixed answer that could be memorized. This ensures that models cannot rely on memorization and must genuinely perform the intended reasoning.
>
> Table 3 show that Qwen2.5 models achieve very high scores on realistic tasks, but their performance drops dramatically (often by 50-80%) on synthetic tasks that require the same multi-hop reasoning but cannot be solved by memorization. This indicates that the realistic benchmark is already saturated and mainly reflects memorization, not reasoning. In contrast, synthetic tasks allow us to control for prior knowledge and more accurately evaluate the intended reasoning skills.
>
> **Table 3: Performance of Qwen2.5 Models on both realistic (v1) and synthetic (v2) Multi-Needle Reasoning Tasks**
>
> | Task                        | Qwen2.5-7B v1 | v2   | Δ    | Qwen2.5-14B v1 | v2   | Δ    | Qwen2.5-32B v1 | v2   | Δ    | Qwen2.5-72B v1 | v2   | Δ    |
> | ----------------------------- | --------------- | ------ | ------- | ---------------- | ------ | ------- | ---------------- | ------ | ------- | ---------------- | ------ | ------- |
> | Multi-Needle-Reasoning      | 88.5          | 16.9 | -71.6 | 93.2           | 26.1 | -67.1 | 93.3           | 35.9 | -57.5 | 94.3           | 43.9 | -50.4 |
> | Multi-Needle-Reasoning (EN) | 86.7          | 23.1 | -63.6 | 93.0           | 22.9 | -70.0 | 94.4           | 39.5 | -54.9 | 94.2           | 51.0 | -43.1 |
> | Multi-Needle-Reasoning (ZH) | 90.3          | 10.7 | -79.7 | 93.4           | 29.2 | -64.2 | 92.2           | 32.2 | -60.0 | 94.5           | 36.8 | -57.7 |
> | 2-Needle (EN)               | 89.3          | 44.4 | -44.8 | 96.7           | 44.3 | -52.3 | 96.4           | 73.2 | -23.2 | 98.9           | 79.5 | -19.4 |
> | 2-Needle (ZH)               | 93.3          | 25.7 | -67.6 | 99.9           | 75.9 | -24.0 | 99.3           | 89.5 | -9.8  | 98.8           | 85.5 | -13.3 |
> | 3-Needle (EN)               | 86.3          | 26.6 | -59.7 | 90.1           | 20.2 | -69.8 | 88.7           | 34.7 | -54.0 | 89.2           | 47.8 | -41.4 |
> | 3-Needle (ZH)               | 95.6          | 7.6  | -88.0 | 90.2           | 19.0 | -71.2 | 89.5           | 20.8 | -68.8 | 98.0           | 34.4 | -63.5 |
> | 4-Needle (EN)               | 82.9          | 11.0 | -71.9 | 87.9           | 18.5 | -69.4 | 93.4           | 26.5 | -66.9 | 89.0           | 46.9 | -42.1 |
> | 4-Needle (ZH)               | 94.0          | 6.0  | -88.0 | 98.3           | 15.1 | -83.2 | 97.3           | 14.7 | -82.6 | 98.0           | 17.3 | -80.7 |
> | 5-Needle (EN)               | 88.2          | 10.4 | -77.8 | 97.3           | 8.8  | -88.5 | 99.3           | 23.9 | -75.4 | 99.5           | 29.9 | -69.7 |
> | 5-Needle (ZH)               | 78.5          | 3.4  | -75.1 | 85.2           | 6.8  | -78.4 | 82.7           | 3.8  | -79.0 | 83.4           | 10.0 | -73.4 |
>
> v1: realistic tasks (Wikipedia-based), v2: synthetic tasks. Δ = v2 - v1.
>
> In summary, we believe that synthetic tasks, though less “realistic” in content, are necessary for a valid and robust evaluation of reasoning ability, as they avoid the confounding effect of memorization and better reflect the true capabilities of large language models.

---

> ### Author Response · Authors · 2025-07-14
>
> 2. **Data Generation Process**
>
> Thank you for highlighting the need for clarity regarding the data generation process. Our data generation involves a Python-based procedural approach, where **we programmatically create structured familial relationships ​**using predefined templates randomly combined into synthetic prompts. Each prompt has a corresponding predefined answer to ensure consistency. This process is visually illustrated in our current Figure 1. We have also provided a detailed algorithmic description of the data generation process **in Appendix E** of the revised manuscript, where readers can find a comprehensive explanation. Additionally, we will open-source our code to ensure transparency and reproducibility once the paper is accepted.
>
> 3. **Benchmark Comparison and Metrics**
>
> We acknowledge that concurrent works such as Ruler, LongBench v2, and MRCR have emerged and share a similar goal of evaluating long-context capabilities. While there may be some overlapping aspects among these benchmarks, each work focuses on ​**different perspectives and challenges**​. For example, Ruler mainly investigates information-sparse settings by adding a large amount of irrelevant content, while LongBench v2 emphasizes more realistic, real-world long document tasks such as summarization and question answering, where models may not need to read the entire document. MRCR[2] can be seen as a variant of our multi-needle reasoning task.
>
> In contrast, our work is particularly concerned with whether models can **truly perform retrieval and reasoning over the entire context** in information-dense settings, where the answer genuinely requires reading and understanding the full document. We gradually reduced irrelevant information in our tasks to make the setting more information-dense, which is a key difference from Ruler. When designing the tasks, we used programmatic construction to prevent the model's prior knowledge from affecting the evaluation of retrieval and reasoning abilities, which is also different from LongBench v2.
>
> As Reviewer LtVj pointed out,we also proposed a novel metric, ENL-50 to better measure the model’s actual reasoning ability. We believe our work is complementary to these concurrent benchmarks: while all focus on long-context evaluation, our emphasis is on information-dense scenarios and step-by-step reasoning, providing a unique perspective and metric for the community.
>
> 4. **Clarity of Figure 1**
>
> We acknowledge that Figure 1 is cluttered and that there were some formatting issues when converting to PDF in the previous submission. We sincerely accept this feedback. However, splitting a large figure into multiple smaller ones and redoing the layout would be a little complex and may affect consistency. Therefore, instead of dividing the figure, we have **reorganized the formatting of Figure 1, added more spacing, and reduced some textual content** to improve its readability and make it less crowded in the revised manuscript. We appreciate the suggestion.
>
> 5. **Instruction Truncation Explanation**
>
> "Instruction truncation" occurs when the essential prompt instructions placed at the end of the context become incomplete due to tokenizer differences among models. Since different tokenizers may tokenize the same prompt differently, some models may inadvertently truncate crucial instructions. We resolve this by subtracting a buffer from the target context length during prompt generation, ensuring that all models consistently see the complete instructions. We have also revised this point in Section 3.1 of the updated manuscript.
>
> Thank you again for your thoughtful feedback, which significantly improves the manuscript's clarity and quality.
>
> [1] Inoue, N., Stenetorp, P., & Inui, K. (2019). R4C: A benchmark for evaluating RC systems to get the right answer for the right reason. arXiv preprint arXiv:1910.04601.
>
> [2] Vodrahalli, K., Ontanon, S., Tripuraneni, N., Xu, K., Jain, S., Shivanna, R., ... & Olszewska, K. (2024). Michelangelo: Long context evaluations beyond haystacks via latent structure queries. ​*arXiv preprint arXiv:2409.12640*​.

---

### Review · Reviewer_LtVj · 2025-07-01

**Summary Of Contributions:**

This paper introduces NeedleBench, a dataset framework for evaluating Large Language Models (LLMs) on both information-dense and information-sparse tasks in long-context scenarios. Information-sparse tasks include retrieval (single-needle and multi-needle) and reasoning (multi-needle) challenges, while information-dense tasks assess aspects such as name diversity, relationship diversity (across multiple types of relationships between entities), task diversity, language diversity (including English and Chinese), and logical complexity, as well as context length variability. To evaluate LLM performance on information-sparse tasks, the paper proposes a keyword-aware scoring method based on Exact Match (EM), focusing on the accuracy of keyword retrieval and reasoning. For information-dense tasks, Exact Match is also used to measure the model's ability to produce correct answers in the required format. Experiments are conducted with a variety of LLMs, ranging from small to large models, and conclusions are drawn regarding the impact of model size and architecture on performance.

**Audience:**

Yes

**Broader Impact Concerns:**

No broader impact concerns.

**Claims And Evidence:**

Yes

**Requested Changes:**

- Please add detailed results (as an appendix) regarding each experimental dimension, before summarizing/averaging.

**Strengths And Weaknesses:**

Strengths:
- The paper presents a framework for evaluating LLMs on both information-sparse and information-dense tasks, where critical pieces of knowledge (i.e., "needles") may be distributed across different parts of the text. The LLM must then retrieve and/or reason to provide the correct answer. This is an important and relevant problem, and the evaluation methodology proposed by the authors—particularly for information-dense tasks—appears to be novel, to the best of my knowledge.
- The evaluations conducted are thorough and robust, offering sufficient evidence to support the conclusions drawn regarding the performance of LLMs on both information-sparse and information-dense tasks. These evaluations span a range of models with varying sizes and architectures.
- The paper presents valuable insights, particularly in the realm of information-dense tasks. For instance, it finds that small models can only generalize to 2-step reasoning, medium models can handle up to 4 steps, and large models can manage up to 8 steps. Additionally, the identification and measurement of issues like "under-thinking" and other error types in LLMs provides new and useful insights, to the best of my knowledge.

Weaknesses:
- While I appreciate that the paper provides an overall score to evaluate model performance, there are no detailed results (even in an appendix) showing how each model performs on specific tasks. For instance, the performance on information-sparse tasks (e.g., as measured in Eq. 2) and the individual results for information-dense tasks before averaging would have been valuable.
- Regarding the evaluation metric for information-sparse tasks, which focuses on keyword matching, this seems to emphasize retrieval rather than reasoning. Could alternative reasoning metrics be used to better capture the models' ability to reason across these tasks? (e.g. but not limited to [1]).
- For information-sparse tasks, many of the results seem somewhat expected. For instance, it is not surprising that more recent models with global attention mechanisms perform better, that larger models outperform smaller ones in long-context reasoning, or that performance declines as reasoning complexity increases. Additionally, the inclusion of bilingual evaluation does not seem to provide significant new insights into model behavior. Given this, I believe the paper’s most significant contribution lies in the evaluation of information-dense tasks, while the results for information-sparse tasks are largely predictable.

References:
[1] Mondorf, P., & Plank, B. (2024). Beyond Accuracy: Evaluating the Reasoning Behavior of Large Language Models--A Survey. arXiv preprint arXiv:2404.01869.

---

> ### Author Response · Authors · 2025-07-14
>
> Thank you very much for your insightful comments and suggestions. Here are our responses:
>
> 1. **Detailed Results for Each Experimental Dimension**
>
> Thank you for your suggestion regarding more granular results. In response, we have included additional detailed results in Appendix C of the revised manuscript, which now contains results for the Multi-Needle Reasoning sub-tasks (including 2-needle, 3-needle, 4-needle, and 5-needle cases). These supplementary results in the appendix provide a more fine-grained view of model performance.
>
> To avoid overwhelming the appendix with the full experimental space (i.e., every model, context length, needle depth, needle number, and task type), we will make all raw data—including prompts, model outputs, evaluation results, as well as visualizations of model performance—publicly available in a GitHub repository once the paper is accepted. This will ensure transparency and allow interested readers to explore the complete set of results in greater detail.
>
> 2. **On the Use of Keyword Matching Metrics for Information-Sparse Tasks**
>
> In our benchmark, for retrieval-only tasks, we use keyword matching for each information point. For reasoning tasks (e.g., Multi-Needle Reasoning and ATC), we **only match the final answer’s keyword** in the answer box. The model must retrieve all relevant information and reason to produce the single correct answer in the box. This setup ensures the metric truly reflects reasoning ability, as only the final answer is accepted and ​**intermediate keywords are not counted**​. We have also revised and clarified the relevant description in Section 3.1 as suggested. Thank you for your valuable feedback.
>
> 3. **Main Finding: Challenges in Information-Dense Tasks**
>
> We agree with the reviewer that the evaluation of information-dense tasks is indeed the main highlight and contribution of our work. As the reviewer pointed out, the results on information-sparse tasks are largely expected. However, our benchmark reveals that even state-of-the-art reasoning models **still struggle with continuous retrieval and reasoning** in information-dense settings, even when the reasoning patterns are simple. We appreciate the reviewer’s insight on this matter.
>
> Regarding the bilingual evaluation, we acknowledge the reviewer's point that it may not provide dramatically new insights into model behavior. As discussed in Section 4.1.5, our inclusion of bilingual evaluation aimed to ​**enhance accessibility, audience diversity**​, and adoption across linguistically diverse research communities.
>
> Thank you again for your thoughtful feedback.

---

### Decision · Action_Editor_eDdx · 2025-08-18

**Recommendation:** Accept with minor revision

**Additional Comments:**

Please incorporate all the feedback from the reviewers in the final versions.

- Detailed experimental results, e.g. onclude fine-grained results for different sub-tasks, report per-task results (not only averaged scores), etc. Also include results from more models (like DeepSeek, o3, etc) to make the benchmark more comprehensive.
- Expand related work section to include concurrent benchmarks (e.g. Ruler, LongBench v2, MRCR, etc.) and position the paper accordingly.
- Improve clarity (algorithm description, figures, etc.)

**Audience:**

Yes

**Audience Explanation:**

Definitely interesting to people evaluating long context and reasoning models. The benchmark is well-structured, multi-lingual, and highlights critical limitations in state-of-the-art models.

**Claims And Evidence:**

Yes

**Claims Explanation:**

Introduces a benchmark for evaluating LLM retrieval and reasoning across varying information densities. Provides a comprehensive baseline via systematic experiments across multiple models, with detailed results on both information-sparse and information-dense tasks. The experiment design was robust, some concerns about how well the synthetic tasks correlate with real world tasks, but still important addition to the literature on reasoning over long contexts in LLMs.

---

> ### Author Response · Authors · 2025-09-02
>
> Dear Action Editor and Reviewers,
>
> Thank you for your constructive feedback. We have submitted the final camera-ready manuscript, which incorporates all suggested revisions. As promised, the full evaluation results are now publicly available on Hugging Face:
>
> https://huggingface.co/datasets/opencompass/NeedleBench/resolve/main/tmlr_release/needlebench_eval_results.tar.gz
>
> We appreciate your support in bringing this work to publication.

---

> > ### Comment · Action_Editor_eDdx · 2025-09-16
> >
> > Thanks for uploading the revised manuscript with the full evaluation results.
> >
> > However, as reviewer suggested please expand related work section to include concurrent benchmarks (e.g. Ruler, LongBench v2, MRCR, etc.) and position the paper accordingly.

---

> > > ### Author Response · Authors · 2025-09-16
> > >
> > > Dear Action Editor,
> > >
> > > Thank you for your guidance. We have uploaded the revised manuscript.
> > >
> > > As requested, we have expanded the "Long-Context Benchmarks" paragraph in the Related Work section. The new version now explicitly discusses concurrent benchmarks (Ruler, LongBench v2, and MRCR) and clearly positions our work by detailing the trade-offs in current benchmark design and how NeedleBench addresses them.
> > >
> > > We believe this revision incorporates reviewers' suggestions. Thank you again for your support.